# Unfair competition governs the interaction of pCPI-17 with myosin phosphatase (PP1-MYPT1)

Joshua J Filter[1], Byron C Williams[1], Masumi Eto[2], David Shalloway[1], Michael L Goldberg[1]*

[1]Department of Molecular Biology and Genetics, Cornell University, Ithaca, United States; [2]Department of Molecular Physiology and Biophysics, Sidney Kimmel Medical College at Thomas Jefferson University, Philadelphia, United States

**Abstract** The small phosphoprotein pCPI-17 inhibits myosin light-chain phosphatase (MLCP). Current models postulate that during muscle relaxation, phosphatases other than MLCP dephosphorylate and inactivate pCPI-17 to restore MLCP activity. We show here that such hypotheses are insufficient to account for the observed rapidity of pCPI-17 inactivation in mammalian smooth muscles. Instead, MLCP itself is the critical enzyme for pCPI-17 dephosphorylation. We call the mutual sequestration mechanism through which pCPI-17 and MLCP interact *inhibition by unfair competition:* MLCP protects pCPI-17 from other phosphatases, while pCPI-17 blocks other substrates from MLCP's active site. MLCP dephosphorylates pCPI-17 at a slow rate that is, nonetheless, both sufficient and necessary to explain the speed of pCPI-17 dephosphorylation and the consequent MLCP activation during muscle relaxation.

*For correspondence: mlg11@
cornell.edu

**Competing interests:** The
authors declare that no
competing interests exist.

**Reviewing editor:** Roger J
Davis, University of
Massachusetts Medical School,
United States

## Introduction

The phosphorylation of certain small proteins converts these polypeptides into inhibitors of PPP family phosphatases [reviewed by (*Eto and Brautigan, 2012*)]. These phosphoproteins have extremely strong affinities for the holoenzymes they target with $IC_{50}$ values in the nanomolar/ subnanomolar range. Given that the binding of phosphatase and phosphorylated inhibitor is so tight, how can the enzyme's activity be restored quickly when the phosphatase needs to be reactivated?

Current models assume that the phosphorylated inhibitors act as pseudosubstrates that bind to the phosphatase's active site, blocking access of genuine substrates (*Desdouits et al., 1995*; *Eto et al., 2004*; *Wakula et al., 2003*). Either the enzyme cannot dephosphorylate the pseudosubstrate, or the dephosphorylation is too slow to contribute meaningfully to the system's reboot. Thus, in these models, the inhibition can only be relieved by dissociation of the phosphorylated inhibitor from the enzyme and its subsequent dephosphorylation by other phosphatases. We argue that this scenario is incompatible with rapid changes of cellular state, because the very tight binding of the inhibitor to the phosphatase protects the inhibitor from other enzymes. Instead, the inhibited enzyme itself must dephosphorylate the inhibitor at a carefully calibrated rate that is slow enough to maintain inhibition but fast enough to permit speedy physiological responses.

We examine here whether this idea, which we call *inhibition by unfair competition,* might contribute to the rapidity of smooth muscle relaxation that occurs after treatment with vasodilators. We focus on the regulation of myosin light-chain phosphatase (MLCP), an enzyme comprised of a PP1 catalytic subunit and regulatory subunit MYPT1. MLCP targets the myosin regulatory light chain (MRLC) and other proteins including ezrin/radixin/moesin (ERM) to affect cytoskeletal organization in smooth muscle and nonmuscle cells [reviewed in (*Grassie et al., 2011*; *Hartshorne et al., 1998*;

*Ito et al., 2004*)]. Because dephosphorylation of MRLC deactivates myosin motor activity, MLCP is also a critical determinant of smooth muscle force: Agonist-mediated MLCP inhibition augments force development (*Hartshorne et al., 1998*; *Isotani et al., 2004*; *Ito et al., 2004*; *Somlyo and Somlyo, 2003*), while nitric oxide simulation causes rapid MLCP re-activation concurrent with smooth muscle relaxation (*Etter et al., 2001*; *Lubomirov et al., 2006*; *Somlyo and Somlyo, 2003*). Genetic experiments have validated the indispensability of MLCP: Ablation of MYPT1 in mice causes embryonic lethality, while conditional knockout of MYPT1 in smooth muscles slows vasodilation and elevates blood pressure [reviewed in (*Grassie et al., 2012*; *Hartshorne et al., 1998*; *Ito et al., 2004*; *Qiao et al., 2014*)].

Smooth muscle, neurons, and other cells contain a small endogenous regulatory protein specific to MLCP, named CPI-17 (*Eto et al., 1995*, *1997*). This 17 kDa polypeptide becomes an MLCP inhibitor when it is phosphorylated at $Thr^{38}$ by any of several kinases, including PKC and ROCK, that are activated when smooth muscle is exposed to agonists [reviewed in (*Eto, 2009*; *Eto and Brautigan, 2012*)]. pCPI-17 binds tightly to MLCP with $pThr^{38}$ occupying the enzyme's active site (*Eto, 2009*; *Eto et al., 2007*; *Hayashi et al., 2001*), thereby inactivating MLCP and increasing both MRLC phosphorylation and tissue contractility (*Li et al., 1998*). CPI-17 $Thr^{38}$ and MRLC phosphorylation levels coordinately correspond with smooth muscle contraction during many physiological processes within smooth muscles and other cell types [e.g. (*Deng et al., 2002*; *Eto et al., 2002*; *Li et al., 1998*; *Niiro et al., 2003*; *Watanabe et al., 2001*)].

Our concern here is the dephosphorylation of CPI-17 and reactivation of MLCP, which occur in parallel with rapid relaxation of smooth muscles within seconds of nitric oxide stimulation (*Etter et al., 2001*; *Kitazawa et al., 2003*, *2009*). Previous investigators have assumed that hydrolysis of pCPI-17 by MLCP is negligible and therefore envision that pCPI-17 must first dissociate from MLCP to allow other enzymes (hereafter called 'PPU' for 'Protein Phosphatase Unknown') to then rapidly dephosphorylate the freed inhibitor. Some investigators believe that these PPUs are PP1-containing holoenzymes different from MLCP (*Eto and Brautigan, 2012*; *Eto et al., 2004*; *Kitazawa, 2010*), but others concluded that the CPI-17 dephosphorylating enzymes are PP2A (*Hersch et al., 2004*; *Obara et al., 2010*; *Takizawa et al., 2002*) and/or PP2C (*Takizawa et al., 2002*). However, regardless of the identity of PPU, these previous models do not adequately consider the consequences of the very tight binding of pCPI-17 to MLCP, which some experiments suggest is subnanomolar (*Eto et al., 1995*, *1997*, *2000*). This binding could protect pCPI-17 from any PPU enzymes, implying that MLCP reactivation must depend on its own dephosphorylation of its own inhibitor by the unfair competition principle.

Evaluating this possibility requires careful quantitative analysis. To this end, we have measured the key parameters including the pCPI-17/MLCP and pCPI-17/PPU dephosphorylation kinetic constants, the effective PPU concentration (which we resolve into two separate components), and the effect of competing substrates on the dephosphorylation of pCPI-17 by PPU and MLCP. We use these and prior experimental values to model quantitatively the time-courses of pCPI-17 dephosphorylation and MLCP reactivation during vasodilation. We show that, because MLCP effectively protects pCPI-17 against other phosphatases, only models that include MLCP's dominating dephosphorylation of pCPI-17 can explain the experimental physiological measurements (*Kitazawa et al., 2009*).

We have previously shown that a similar unfair competition mechanism is important for the release of PP2A-B55 from inhibition by phosphorylated Endosulfine (pEndos) during exit from mitosis (*Williams et al., 2014*). The fact that this mechanism operates to control two different phosphatases of the PPP family [PP2A-B55 and PP1-MYPT1 (MLCP)] suggests that it is ancient and evolved prior to the duplication and divergence of PPP phosphatase genes in early eukaryotes.

## Results

### Current models for pCPI-17 inactivation are unlikely to account for the rapidity of smooth muscle relaxation

Adding the vasodilator sodium nitroprusside to contracted rabbit femoral artery leads to maximal dephosphorylation of pCPI-17 with a half-life $t_{1/2} \sim 10$ s, resulting in near-simultaneous phosphorylation of the myosin regulatory light chain and a decrease in contractile force soon thereafter

(**Kitazawa et al., 2009**). As described in the Introduction, current models posit that pCPI-17 dephosphorylation is accomplished solely by PPU enzymes. Is this assumption consistent with the observed speed of the process? We argue that the answer is likely 'no' under physiological conditions.

**Figure 1A** shows the fastest mechanism by which such models could operate: PPU enzymes immediately dephosphorylate any pCPI-17 that dissociates from MLCP, and the dephosphorylated CPI-17 is not rephosphorylated. The rate-limiting step of this scenario is simply the off-rate ($k_{off}$) for the dissociation of pCPI-17 from MLCP; that is, $k_{off}$ = log $2/t_{1/2}$ ~ 0.07 s$^{-1}$. However, this model is an oversimplification that ignores two important counteracting processes: rebinding of released pCPI-17 and regeneration of pCPI-17 from CPI-17.

These processes are likely to be important because prior experiments, combined with the lower bound on $k_{off}$ required by the physiological rate, indicate that the pCPI-17/MLCP association constant, $k_{on}$, must be very high. Reported measurements for the IC$_{50}$ of the inhibition of MLCP by thio-phosphorylated CPI-17 [thio(p)CPI-17], which cannot be dephosphorylated, fall within the range of 0.12–6 nM (**Erdődi et al., 2003**; **Eto et al., 1995**, **1997**, **2000**, **2004**; **Hayashi et al., 2001**). Because, for tight-binding inhibitors, the $K_i$ is lower than the IC$_{50}$ (see **Supplementary file 1**), we can estimate that $k_{on}$ (= $k_{off}/K_i$) is minimally 1.2 × 10$^7$ M$^{-1}$ s$^{-1}$ and would need to be more than 6 × 10$^8$ M$^{-1}$ s$^{-1}$ if the $K_i$ were as low as 0.12 nM. Such association rates are very fast: In a recent tabulation of the kinetic parameters of 144 protein complexes, only two had $k_{on}$ values in excess of 1 × 10$^8$ M$^{-1}$ s$^{-1}$ (**Archakov et al., 2003**; **Moal and Bates, 2012**). In contrast, the corresponding association constants for the potential PPU enzymes are probably less than 10$^5$ M$^{-1}$ s$^{-1}$ (**McWhirter et al., 2008**; **Pan et al., 2015**; **Price and Mumby, 2000**). Therefore, unless the concentration of accessible PPU is orders of magnitude greater than the concentration of unbound MLCP, rebinding will strongly retard dephosphorylation by PPU.

Fast rebinding would also sequester pCPI-17 regenerated from CPI-17 by kinases such as PKC and ROCK, which likely retain residual activity during smooth muscle relaxation. Such residual kinase activity is particularly likely when vasodilators are added in the presence of agonists such as histamine (**Etter et al., 2001**) or phenylephrine (**Kitazawa et al., 2009**; **Shibata et al., 2015**). As one example, two indicators of ROCK activity (pRhoA Ser$^{188}$ and pMYPT1 Thr$^{853}$) remained almost unchanged after vasodilator administration even when pCPI-17 decreased to a minimum level (**Kitazawa et al., 2009**).

**Figure 1B** presents a more realistic model that accounts for these counteracting processes. We show below that the $K_m$ for pCPI-17 lies at the lower end of the reported thio(p)CPI-17 IC$_{50}$ range and is ~0.5 nM. Thus, even in this model, $k_{off}$ cannot be much higher than 0.07 s$^{-1}$ because the requisite $k_{on}$ [= ($k_{off}$ + $k_{cat}$)/$K_m$] would be unrealistically high. The net rates at which pCPI-17 is dephosphorylated and free MLCP is created therefore must be slower than for the model in **Figure 1A** to a degree that depends on the relative concentrations and kinetic properties of all the players (CPI-17, MLCP, PPU, and PKC/ROCK). We present below a mathematical treatment of the **Figure 1B** model using values for these parameters determined from the literature and from our own experiments. For the time being, we emphasize simply that the actual rate of MLCP reactivation observed experimentally [$t_{1/2}$ ≈ 10 s; (**Kitazawa et al., 2009**)] is approximately 10-fold faster than the rate achievable in cells through the **Figure 1B** model.

## MLCP can dephosphorylate pCPI-17 at a physiologically relevant rate

The discussion above suggests that the rapid reactivation of MLCP requires some method to rid the enzyme of the pCPI-17 inhibitor other than simple dissociation. The obvious strategy would be for MLCP itself to dephosphorylate pCPI-17, as indicated in **Figure 1C**. This idea is plausible because pThr$^{38}$ of MLCP- bound pCPI-17 is located at the enzyme's active site (**Eto, 2009**; **Eto et al., 2007**).

However, because it is small, it has been difficult to measure the rate of MLCP-mediated pCPI-17 dephosphorylation, and previous reports have only been able to place an upper bound (<0.1 s$^{-1}$) on its value (**Hayashi et al., 2001**). A $k_{cat}$ toward the upper limit of this range would be consistent with the model in **Figure 1C**, but a significantly smaller value would not. Therefore, it was important to determine if MLCP can dephosphorylate pCPI-17 at a rate fast enough to account for its dephosphorylation during muscle relaxation.

To this end, we prepared MLCP by transiently transfecting HEK-293 CRL-1573 cells with a construct overexpressing FLAG-tagged MYPT1. **Figure 2** shows that a complex containing PP1$\beta$/$\delta$ (PPP1CB) and MYPT1 was obtained. This preparation lacks a small M20 subunit that is found in

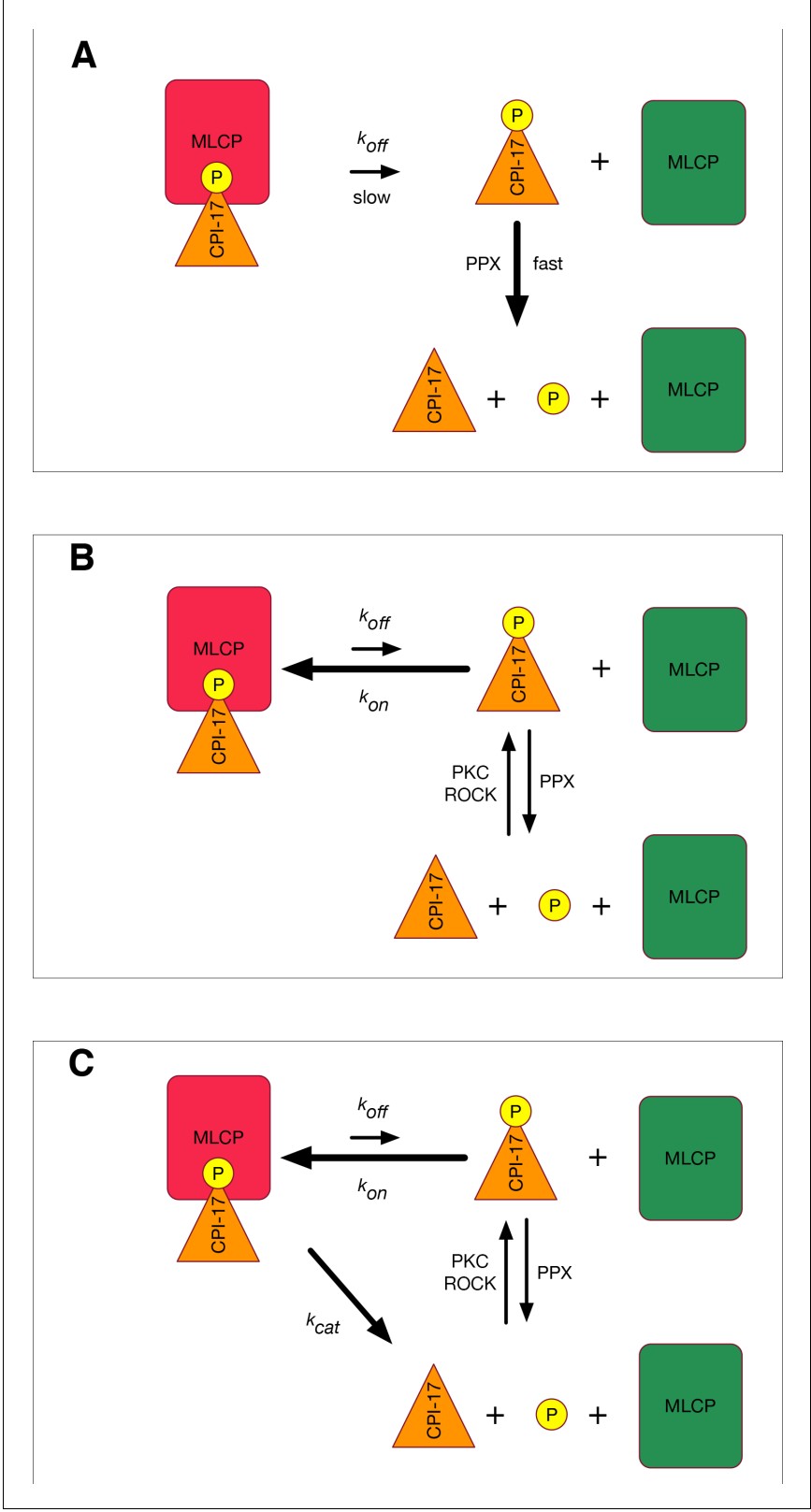

**Figure 1.** Models for MLCP reactivation. (**A**) A simplistic model in which pCPI-17 is dephosphorylated only by other phosphatases (PPU) immediately after it dissociates from MLCP. The rate-limiting step for MLCP reactivation is the dissociation, whose rate is $k_{off}$. (**B**) A more realistic scenario in which pCPI-17 freed by dissociation can rebind to free MLCP, and in which pCPI-17 is in an equilibrium with unphosphorylated CPI-17 that depends on the

*Figure 1 continued on next page*

*Figure 1 continued*

relative activities of the kinases (PKC) and phosphatases (PPU) involved. The rate of MLCP inactivation must necessarily be slower than that in (**A**). (**C**) Unfair competition. pCPI-17 is removed from MLCP by dephosphorylation as well as by dissociation. Free pCPI-17 and unphosphorylated CPI-17 are in an equilibrium determined by the relative activities of the kinases (PKC) and phosphatases (MLCP and PPU) involved. When the concentration of pCPI-17 is less than or equal to the concentration of MLCP, the very low $K_m$ ensures that almost all pCPI-17 dephosphorylation is accomplished by MLCP.

MLCP purified from tissues but whose role has never been clarified (*Ito et al., 2004*), so we performed multiple tests of the enzyme's functional authenticity. Consistent with the known activities of MLCP, we found that the complex we purified is able to dephosphorylate the myosin regulatory light chain (pMRLC); the C terminal domain (pC-ERMAD) of ERM proteins, which are physiological MLCP substrates (*Eto et al., 2005*, *2000*; *Fukata et al., 1998*)]; and the nonspecific substrate myelin basic protein (pMyBP). The assays with all three substrates (*Figure 2—figure supplement 1A–C*) were linear with respect to time and enzyme concentrations under the experimental conditions used [notably, in the absence of $Mn^{2+}$, which can induce artifactual activities of PPP phosphatases (*Watanabe et al., 2003*)]. Moreover, as shown below, dephosphorylation of all three substrates was inhibited by the drugs okadaic acid and calyculin A, with $IC_{50}$ values in the identical range of published studies. The purified recombinant complex (*Figure 2A*) thus has properties very similar to

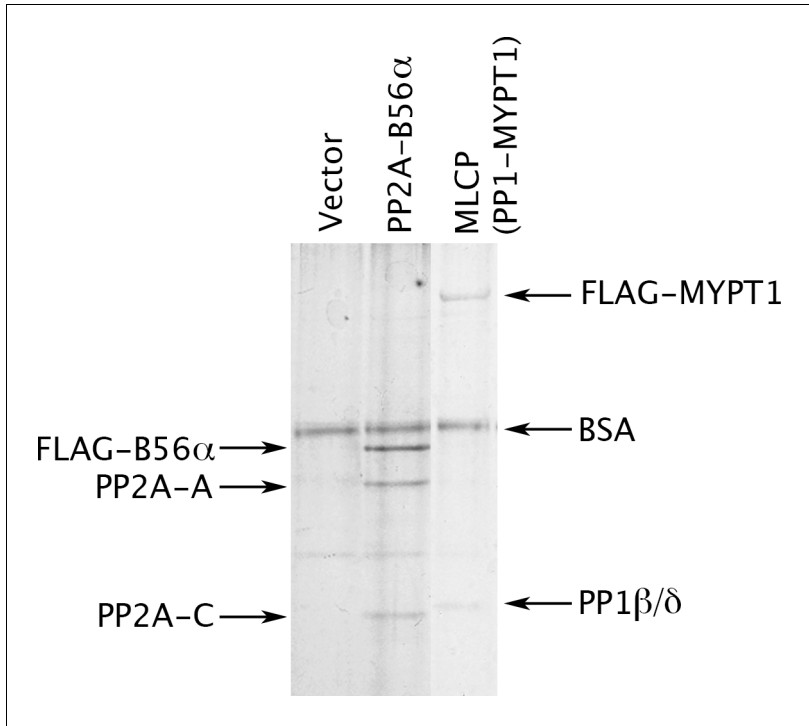

**Figure 2.** Preparation of MLCP. HEK-293 CRL-1573 cells were transformed with empty vector, with a construct overexpressing a FLAG-tagged version of the B56α regulatory subunit of PP2A for comparison, or with a construct overexpressing FLAG-MYPT1. Phosphatase holoenzymes were then purified as previously described (*Adams and Wadzinski, 2007*; *Williams et al., 2014*) and stabilized by the addition of bovine serum albumin (BSA). The myosin phosphatase holoenzyme in the third lane has both FLAG-MYPT1 and PP1β/δ in roughly equal stoichiometry; the identity of these proteins was verified by Western blotting with anti-FLAG and with antibodies specifically directed against the α, β/δ, and γ isoforms of the PP1 catalytic subunit (data not shown).

The following figure supplement is available for figure 2:

**Figure supplement 1.** Phosphatase activities of the MLCP preparation shown in *Figure 2*.

those of authentic MLCP. Another study recently reached the same conclusion for PP1$\beta$/$\delta$-MYPT1 complex purified in substantially the same way; importantly, the presence or absence of the M20 subunit had no obvious effect on the activities of the complex (*Khasnis et al., 2014*).

Wild-type CPI-17 contains two sites that can be phosphorylated by PKC (*Eto et al., 1995*), Thr[38] (the major phosphosite that converts CPI-17 into an MLCP inhibitor) and the minor site Ser[12] [which lies outside the inhibitory domain and regulates the protein's nuclear import; (*Eto et al., 2013*)]. So that we could quantitate accurately phosphorylation/dephosphorylation at the single Thr[38] site, we created recombinant CPI-17 with a Ser[12]-to-Ala[12] mutation. The Ala[12] mutant behaved identically to the wild-type protein as an MLCP inhibitor (see Materials and methods). *Figure 2—figure supplement 1F* shows that this recombinant MLCP also dephosphorylates pCPI-17, although quite slowly, with a catalytic rate $k_{cat}$ ~0.08 s[−1]. Combining this measurement with values obtained in the additional experiments described below (*Figures 3*, *4* and *5*) gave a mean estimate of $k_{cat}$ = 0.06 ± 0.01 s[−1] (n = 4) at 30°C, which was used for subsequent analyses (*Table 1*). This value for the dephosphorylation of pCPI-17 by MLCP is more than 100-fold lower than the $k_{cat}$ of 10.7–13.7 s[−1] previously measured by other investigators for the dephosphorylation of the myosin regulatory light chain (MRLC) by the same enzyme (*Feng et al., 1999a*; *Ichikawa et al., 1996*), and it is also very slow compared with the action of other phosphatases on pCPI-17 [(*Eto et al., 2004*; *Hersch et al., 2004*; *Takizawa et al., 2002*) and see below]. However, it is fast enough to account for the physiological dephosphorylation rate discussed in the Introduction. And indeed, if the $k_{cat}$ was substantially higher than 0.1 s[−1], pCPI-17 would be unable to inhibit MLCP.

## The $K_m$ of the pCPI-17/MLCP interaction is extraordinarily low

Although MLCP dephosphorylates pCI-17 much more slowly than other phosphatases targeting the same substrate [(*Eto et al., 2004*; *Hersch et al., 2004*; *Takizawa et al., 2002*) and see below], their association might be sufficiently tight so as to shield pCPI-17 from any possible PPU phosphatases.

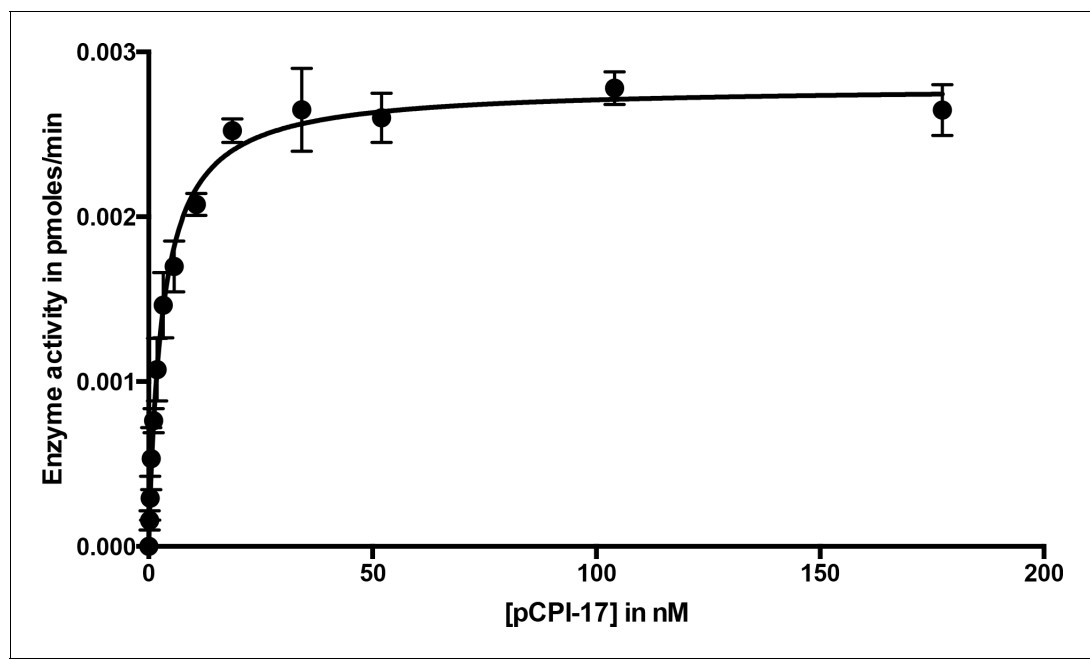

**Figure 3.** Dephosphorylation of pCPI-17 by MLCP as a function of pCPI-17 concentration. All assays contained 0.1 nM MLCP and were performed for 3 min at 30°C in a total volume of 8 µL. The experiment was replicated in triplicate (n = 3); error bars represent standard deviations. Data were analyzed by non-linear regression to Michaelis-Menten kinetics with Prism 6 software (GraphPad Software, Inc., La Jolla, CA). For this experiment, $K_m$ = 3.03 ± 0.22 nM; $k_{cat}$ = 0.059 ± 0.002 s[−1].

The following figure supplement is available for figure 3:

**Figure supplement 1.** Dephosphorylation of pCPI-17 by MLCP as a function of pCPI-17 concentration.

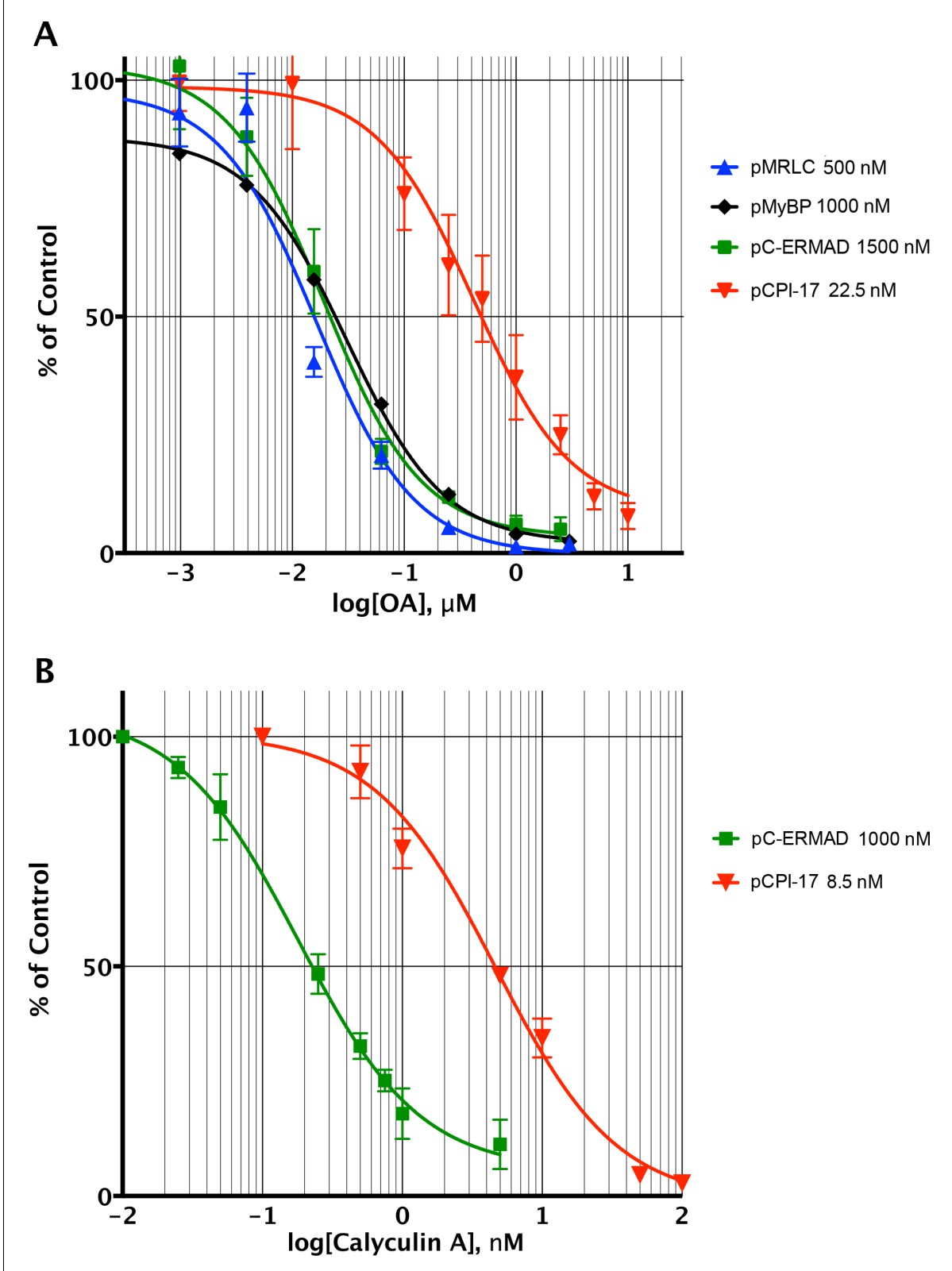

**Figure 4.** Determining the pCPI-17/MLCP $K_m$ by competitive inhibition. (**A**) Inhibition of MLCP by okadaic acid (OA). Enzyme assays in the presence of increasing levels of OA were performed with the indicated substrates at the indicated concentrations; in all assays, the concentration of MLCP was 0.1 nM. Values are presented as a percentage of the average of four control reactions for each substrate without OA. For pCPI-17, each point represents the average, and error bars the standard deviation, of five trials ($n = 5$); for phosphorylated myosin regulatory light chain (pMRLC), $n = 4$; for pC-

*Figure 4 continued on next page*

*Figure 4 continued*

ERMAD, $n$ = 2; for pMyelin Basic Protein (pMyBP), $n$ = 1. Because the MLCP concentration is very low, the $IC_{50}$s of the pMRLC, pMyBP, and pC-ERMAD reactions must be close to the $K_i$ for OA, which is therefore ~20 nM. The ~10–20× larger $IC_{50}$ with pCPI-17 indicates that this substrate can compete effectively with OA because its $K_m$ is smaller than the $K_i$. Indeed, the best-fit $K_m$ determined by nonlinear regression of this and a similar experiment using 8.75 nM pCPI-17 ($n$ = 3; not shown) is 0.59 ± 0.05 nM (see **Supplementary file 1**). (**B**) Inhibition of MLCP by calyculin A. Enzyme assays in the presence of increasing levels of calyculin A were performed with pCPI-17 or with pC-ERMAD at the concentrations indicated; the concentration of MLCP was 0.1 nM. Values are presented as a percentage of the average of three control reactions for each substrate without OA; three replicates of each data point were assayed ($n$ = 3), and error bars represent the standard deviations. The measured $K_i$ of calyculin A for MLCP (from the dephosphorylation of pC-ERMAD) was 0.17 ± 0.05 nM. As with the OA experiments in part (**A**), the increased $IC_{50}$ with pCPI-17 (~30×) reflects competition of this substrate with calyculin; the nonlinear regression best-fit is $K_m$ = 0.36 ± 0.10 nM for the pCPI-17 dephosphorylation reaction.

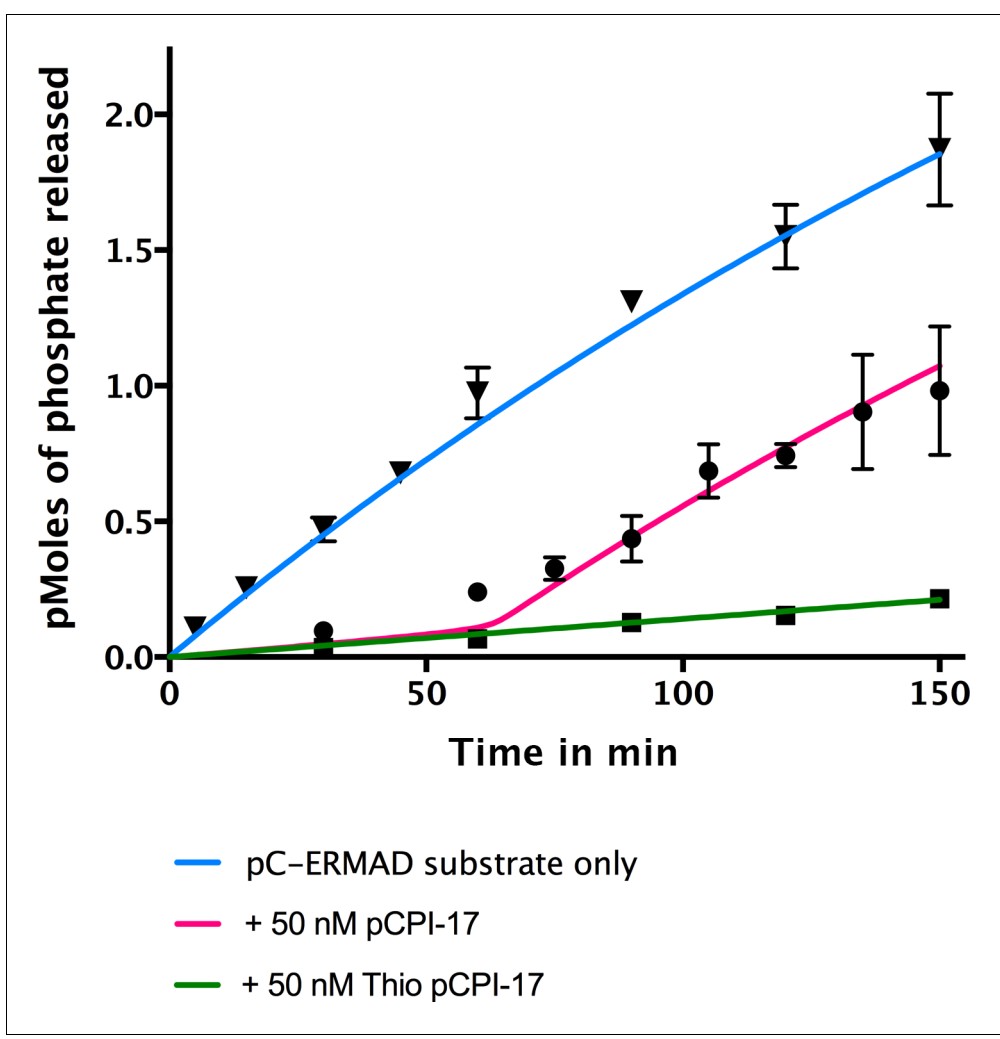

**Figure 5.** In vitro demonstration of the unfair competition mechanism for MLCP and pCPI-17. Enzyme assays containing 0.25 nM MLCP and 1.4 µM C-ERMAD were performed at 30°C for the indicated times, either in the absence of CPI-17 (inverted triangles) or in the presence of 50 nM pCPI-17 (circles) or 50 nM thio(p)CPI-17 (squares). $N$ = 3 for all data points; error bars represent standard errors of the mean. The data show that MLCP must first dephosphorylate pCPI-17 before it can target the C-ERMAD substrate. The blue and green lines are nonlinear regressions to the data in the absence of pCPI-17. The red line is the theoretical prediction based on the previously measured pCPI-17/MLCP kinetic constants as described in **Supplementary file 1**.

**Table 1.** Phosphatase and substrate constants and concentrations.

| Phosphatase | Substrate/Inhibitor | Constant | Value | Source figure or (reference) |
|---|---|---|---|---|
| | pCPI-17 | $([CPI\text{-}17]+[pCPI\text{-}17])_{phys}$ | 7 µM | (**Kitazawa and Kitazawa, 2012**; **Woodsome et al., 2001**) |

| Phosphatase | Substrate/Inhibitor | Constant | Value | Source figure or (reference) |
|---|---|---|---|---|
| MLCP | | $[MLCP]_{phys}$ | 1 µM | (**Alessi et al., 1992**; **Shirazi et al., 1994**) |
| | pCPI-17 | $K_m$ | 0.48 ± 0.03 nM | **Figure 4** |
| | | $k_{cat}$ | 0.06 ± 0.01 s$^{-1}$ | **Figure 2—figure supplement 1**, **Figures 3**, **4** and **5** |
| | OA | $K_i$ | 20 ± 2 nM | **Figure 4** |
| | calyculin A | $K_i$ | 0.13–0.22 nM | **Figure 4** |

| Phosphatase | Substrate/Inhibitor | Constant | Value | Source figure or (reference) |
|---|---|---|---|---|
| PP2A-B56α | pCPI-17 | $K_m$ | 4.2 ± 0.5 µM | **Figure 6** |
| | | $k_{cat}$ | 29 ± 2 s$^{-1}$ | **Figure 6** |

| Phosphatase | Substrate/Inhibitor | Constant | Value | Source figure or (reference) |
|---|---|---|---|---|
| PPU$^a$ | | $[PPU]^a_{phys}$ | ~1.5 µM | **Figure 8** |
| | pCPI-17 | $K_m$ | 15 ± 2 µM | **Figure 7—figure supplement 2** |
| | | $k_{cat}$* | 41 ± 6 s$^{-1}$ | **Figure 7**, **Figure 7—figure supplement 1** and **Figure 8** |
| | OA | $K_i$ | <0.5 nM | **Figure 7** and **Figure 7—figure supplement 1** |

| Phosphatase | Substrate/Inhibitor | Constant | Value | Source figure or (reference) |
|---|---|---|---|---|
| PPU$^b$ | pCPI-17 | $K_m$ | ≥15 µM | **Figure 7—figure supplement 2** |
| | | $k_{cat}\,[PPU]^b_{phys}/K_m$ | ~0.7 s$^{-1}$ | **Figure 7** and **Figure 7—figure supplement 1** |
| | OA | $K_i$ | ~450 nM | **Figure 7** and **Figure 7—figure supplement 1** |

All parameters are at 30°C.

phys: Physiological total (bound plus unbound) concentration; these values are computed using an estimated physiological total protein concentration of 170 mg/ml (**Wiśniewski et al., 2014**).

PPU$^a$ is the PP2A-like component of PPU; it contributes 85% of the pCPI-17-directed PPU phosphatase activity. PPU$^b$ is the minor OA-resistant component.

*Computed from $k_{cat}\,[PPU]^a/M/K_m$, $[PPU]^a/M$, and $K_m$ (see **Table 2**).

For this reason, an accurate measure of the $K_m$ for the dephosphorylation of pCPI-17 by MLCP is critically important. We estimated this value first from classical Michaelis-Menten experiments determining the initial velocity as a function of pCPI-17 concentration; this method indicated the $K_m$ to be ~3 nM (**Figure 3** and **Figure 3—figure supplement 1**). However, this value is likely a considerable overestimate because, in samples with the requisite low concentrations of pCPI-17, substantial proportions of the substrate were depleted during the reaction.

We therefore refined this estimate using a variant of a more accurate method developed for tight-binding reactions (**Takai et al., 1995**; **Williams et al., 2014**) — by measuring the ability of the pCPI-17 dephosphorylation reaction to compete with inhibition by okadaic acid (OA; **Figure 4A**) or by calyculin A (**Figure 4B**) for the active site of MLCP. The first step was to measure accurately the $K_i$s of these inhibitors for MLCP using pMRLC, pC-ERMAD, and pMyBP substrates (blue, green, and black lines, respectively in **Figure 4**). These $K_i$s can be calculated from, and are close to, the IC$_{50}$s because these substrates have large $K_m$s, and thus cannot compete with the inhibitors (see

**Table 2.** Additional phosphatase constants.

| Phosphatase | Substrate | Constant | Value | Source figure or (reference) |
|---|---|---|---|---|
| MLCP | | $\Sigma_{phys}$* | ~15 | *Figure 8—figure supplement 2* |
| | pMyBP | $K_m$ | ~1.7 μM | *Figure 2—figure supplement 1* |
| | | $k_{cat}$ | ~0.18 s$^{-1}$ | *Figure 2—figure supplement 1* |
| | pC-ERMAD | $K_m$ | >2.5 μM | *Figure 2—figure supplement 1* |
| | | $k_{cat}/K_m$ | $2.6 \pm 0.03 \times 10^{-4}$ nM$^{-1}$ s$^{-1}$ | *Figure 2—figure supplement 1* |
| | pMRLC | $K_m$ | ~16 μM | (*Kawano et al., 1999*) |

| Phosphatase | Substrate | Constant | Value | Source figure or (reference) |
|---|---|---|---|---|
| PPU | | $\Sigma_{phys}$* | ~24 | *Figure 8—figure supplement 2* |
| | pCPI-17 | $K_m$ | 15 ± 2 μM | *Figure 7—figure supplement 2* |
| | | $K_m$(0°C) | 14 ± 3 μM | *Figure 7—figure supplement 2* |

| Phosphatase | Substrate | Constant | Value | Source figure or (reference) |
|---|---|---|---|---|
| PPU[a] | pCPI-17 | $k_{cat}$ (30°C)/$k_{cat}$ (0°C) | 8.3 ± 0.8 | *Figure 7—figure supplement 1* |
| | | $k_{cat}$ [PPU][a]/M/$K_m$ | 0.024 ± 0.001 s$^{-1}$ ml mg$^{-1}$ | *Figure 7* and *Figure 7—figure supplement 1* |
| | | [PPU][a]/M | 8.9 ± 0.9 pmol/mg | *Figure 8* |

| Phosphatase | Substrate | Constant | Value | Source figure or (reference) |
|---|---|---|---|---|
| PPU[b] | pCPI-17 | $k_{cat}$ [PPU][b]/M/$K_m$ | 0.0045 ± 0.0002 s$^{-1}$ ml mg$^{-1}$ | *Figure 7* and *Figure 7—figure supplement 1* |

All parameters are at 30°C except as specified.

phys: Physiological total (bound plus unbound) concentration; these values are computed using an estimated physiological total protein concentration of 170 mg/ml (*Wiśniewski et al., 2014*).

M: total extract protein concentration

Σ: Competition factor from other substrates (see *Supplementary file 1*)

*Figure 2—figure supplement 1*, *Table 2*, and *Supplementary file 1*). The experiments gave an OA $K_i$ of 20 ± 2 nM (*n* = 7), which is comparable with many published studies (*Alessi et al., 1992*; *Ishihara et al., 1989*; *Mitsui et al., 1992*; *Shirazi et al., 1994*; *Swingle et al., 2007*); and a calyculin A $K_i$ range of 0.13–0.22 nM (*n* = 4), which is slightly lower than the value previously reported for inhibition of the catalytic subunit of PP1 by this drug [0.4 nM (*Swingle et al., 2007*)]. The second step was to determine the OA and calyculin A IC$_{50}$s for the dephosphorylation of pCPI-17 by MLCP (red lines in *Figure 4*), which were found to be more than one order-of-magnitude higher than those for the other substrates. The reason is that the very high affinity of pCPI-17 for MLCP allows pCPI-17 to compete effectively with the small molecule inhibitors; therefore, higher concentrations of the drugs are needed to inhibit pCPI-17 dephosphorylation. Analysis of the combined results, as described in *Supplementary file 1*, gave $K_m$ = 0.48 ± 0.03 nM [*n* = 8 (OA); *n* = 3 (calyculin A)] for the pCPI-17/ MLCP reaction. This more accurate determination, which is substantially lower than the estimate from the substrate titration, corresponds with values at the low end of the range of previous IC$_{50}$ measurements made using thio(p)CPI-17 [0.12–6 nM (*Erdődi et al., 2003*; *Eto et al., 1995*, *1997*, *2000*, *2004*; *Hayashi et al., 2001*)].

The only precedent for a phosphatase reaction with a subnanomolar $K_m$ is that for the dephosphorylation of pEndos by PP2A-B55, the first example of the unfair competition mechanism [(*Williams et al., 2014*); see Discussion]. In fact, very few if any other enzymatic reactions are reported in the BRENDA enzyme database having $K_m$s in the low nanomolar or even tens of nanomolar ranges (*Schomburg et al., 2013*). Importantly, because to our knowledge all other measured $K_m$s of reactions between phosphatases and their 'normal' substrates are higher than 0.5 μM, we can be certain that the pCPI-17 dephosphorylation observed in our experiments is catalyzed by recombinant MLCP and not by any contaminating phosphatase.

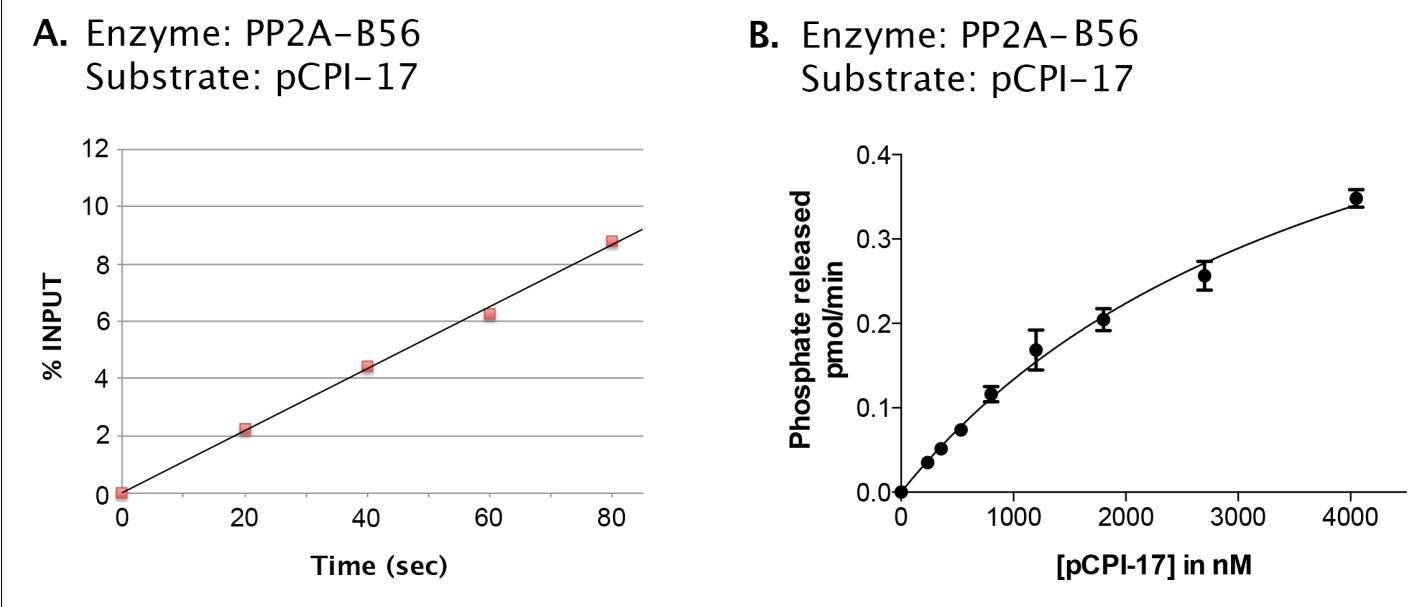

**Figure 6.** Kinetic analysis of the dephosphorylation of pCPI-17 by PP2A-B56α, a candidate PPU enzyme. (**A**). Linearity of assay employing PP2A-B56α enzyme (at 0.25 nM); the concentration of pCPI-17 was 1 μM. (**B**) Velocity of the dephosphorylation of pCPI-17 by 0.1 nM PP2A-B56α as a function of the concentration of the pCPI-17 substrate. Reactions were performed for 1 min at 30°C in a volume of 4 μL. The data were analyzed by Prism 6 software as in *Figure 4*, giving $K_m$ = 4.2 ± 0.5 μM and $k_{cat}$ = 29 ± 2 s$^{-1}$.

## pCPI-17 and MLCP interact through unfair competition

The low $k_{cat}$ and extremely low $K_m$ values for the interaction between MLCP and pCPI-17 dictate that as long as pCPI-17 is available, it will occupy the active site of MLCP and prevent the enzyme from dephosphorylating other substrates. However, when the kinases that phosphorylate pCPI-17 become inactivated, MLCP can dephosphorylate the bound pCPI-17 and the enzyme can then turn its attention to other normal substrates such as the phosphorylated myosin regulatory light chain and ERM proteins.

*Figure 5* presents an in vitro demonstration of this principle. MLCP was incubated with $^{32}$P-labelled pC-ERMAD substrate in the absence of pCPI-17 or in the presence of an excess of either unlabeled pCPI-17 or thio(p)CPI-17. As expected, thio(p)CPI-17's inhibition of MLCP remained constant throughout the period of observation. In contrast, in the presence of pCPI-17, the rate of pC-ERMAD dephosphorylation stayed low at early time points, when pCPI-17 is in excess of MLCP, but it then began to increase, presumably when the concentration of pCPI-17 was reduced to the concentration of MLCP. The red curve in *Figure 5* was calculated theoretically using the unfair competition model and the pCPI-17/MLCP kinetic constants calculated from *Figures 3* and *4* (*Supplementary file 1*). In this example, because pCPI-17 was in >200 fold excess to MLCP, >50 min were required before the observed or calculated rate increase could take place. The good agreement with the data demonstrates the quantitative accuracy of the model.

## MLCP protects pCPI-17 from other phosphatases in mouse uterus extracts

Other investigators have previously shown that cells contain enzymes other than MLCP that are capable of dephosphorylating pCPI-17 (*Eto and Brautigan, 2012*; *Eto et al., 2004*; *Hersch et al., 2004*; *Kitazawa, 2010*; *Obara et al., 2010*; *Takizawa et al., 2002*). We believe that the majority of this PPU activity is due to forms of PP2A-like enzymes, including various isoforms of PP2A, as well as the closely related PP4 and PP6, for two reasons:

First, we verified that at least one PP2A-containing enzyme, PP2A-B56α, indeed efficiently releases phosphate from pCPI-17 (*Figure 6*) with $k_{cat}$ ~30 s$^{-1}$ (*Table 1*). PP2A-B56α thus dephosphorylates pCPI-17 about 500 times faster than does MLCP when the pCPI-17 substrate is

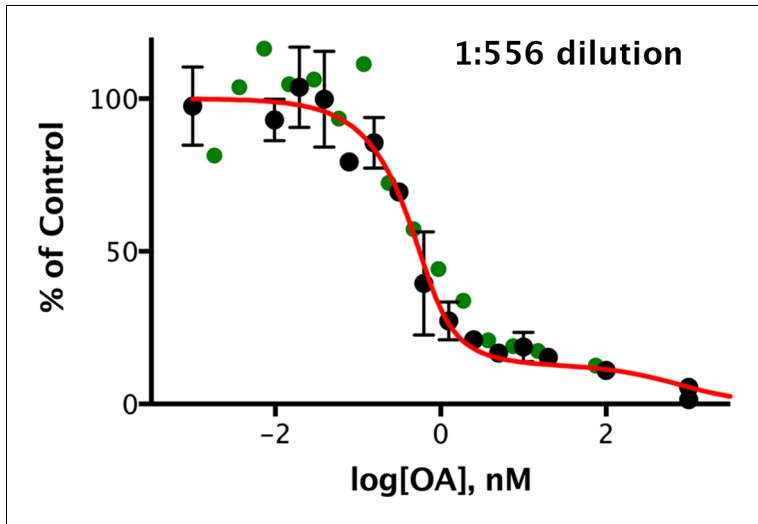

**Figure 7.** Okadaic acid inhibition of the dephosphorylation of pCPI-17 by dilute mouse uterus extracts. Mouse uterus extracts at a dilution of 1:556 were prepared as described in Materials and methods and incubated at 30°C for 45 s with 687 nM $^{32}$P-labeled pCPI-17 plus the indicated concentrations of okadaic acid (OA). The mean phosphate released (with standard deviations; $n = 3$) is shown as a percentage of release in the absence of OA. Since the pCPI-17 concentration is much greater than the MLCP concentration at these dilutions, essentially all dephosphorylation is due to PP2A and other non-MLCP phosphatases (collectively called PPU).

The following figure supplements are available for figure 7:

**Figure supplement 1.** Okadaic acid inhibition of the dephosphorylation of pCPI-17 by dilute mouse uterus extracts.

**Figure supplement 2.** Kinetic analysis of the dephosphorylation of pCPI-17 by PPU.

saturating. However, the Michaelis constant ($K_m$) for the dephosphorylation of pCPI-17 by this PP2A holoenzyme is in the micromolar range, reflecting an affinity for pCPI-17 that is three orders-of-magnitude lower than that of MLCP.

Second, we examined the phosphatase activity directed against pCPI-17 in highly diluted extracts of mouse uterus, a smooth muscle tissue enriched for MLCP (*Aguilar and Mitchell, 2010*). In these experiments (*Figure 7* and *Figure 7—figure supplement 1*), the pCPI-17 concentration was much greater than the estimated MLCP concentration, so we expected that there would be little protection and that most dephosphorylation would be due to PPU enzymes. Quantitative analysis of these data and of the results that will be presented in *Figure 8* below showed that ~85% of the activity was sensitive to subnanomolar OA with $K_m \sim 15 \pm 2$ μM and $k_{cat} = 41 \pm 6$ s$^{-1}$ ($n = 3$), consistent with the known properties of PP2A-like enzymes (*Prickett and Brautigan, 2006*; *Swingle et al., 2007*) and of the purified PP2A holoenzyme example measured in *Figure 6* (see *Supplementary file 1* for the quantitative analysis). We ascribe the residual phosphatase activity (with OA IC$_{50}$ ~ 500 μM) to other PPU phosphatases more resistant to the OA drug.

The observation that the $K_m$s for dephosphorylation of pCPI-17 by PPU enzymes are at least three orders-of-magnitude higher than the $K_m$ for dephosphorylation of pCPI-17 by MLCP (*Table 1*) implies that MLCP will prevent PPU from accessing pCPI-17 when MLCP is in excess of the substrate. In this situation, essentially all pCPI-17 dephosphorylation must be catalyzed by MLCP. Only when pCPI-17 is in excess of MLCP will any appreciable amount of the substrate be available to other phosphatases.

We tested this inference using concentrated mouse uterus extracts that more closely approximate the intracellular situation. These experiments compared the kinetic properties of the phosphatases responsible for pCPI-17 dephosphorylation and their sensitivities to OA-inhibition when the pCPI-17 concentration was either much greater than, or much smaller than, the MLCP concentration

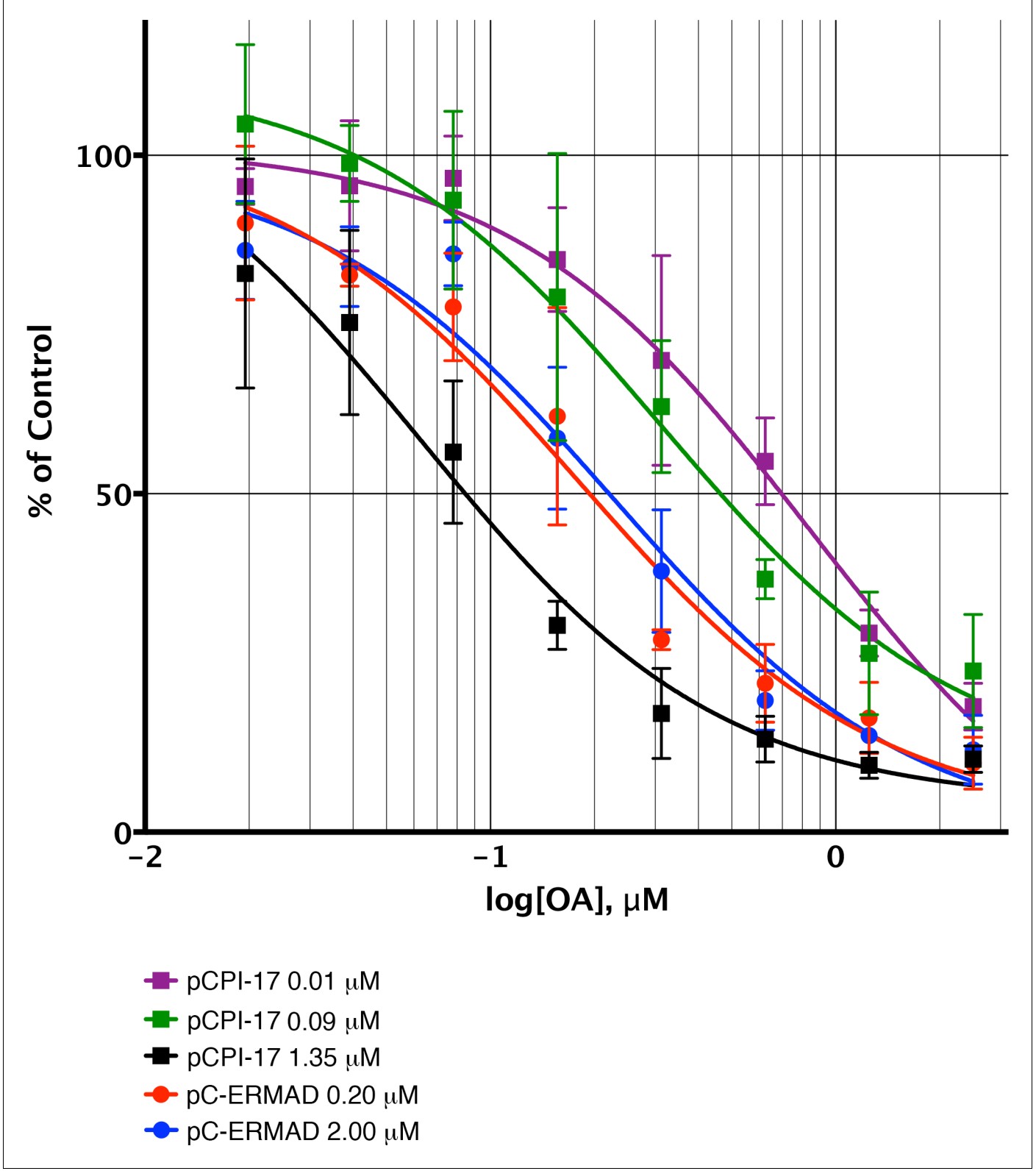

**Figure 8.** Sequestration of pCPI-17 by MLCP in concentrated mouse uterus extracts. Undiluted mouse uterus extracts (~14 mg/ml) were assayed for phosphatase activity using $^{32}$P-labeled pCPI-17 (0°C) or pC-ERMAD (30°C) substrates at the concentrations shown with the indicated concentrations of OA. Results and standard deviations ($n = 3$) are given as a percentage of the average value for each substrate dilution in the absence of OA ($n = 3$). As described in Results and *Supplementary file 1*, the increase in IC$_{50}$ at low pCPI-17 concentrations (i.e. 0.01 or 0.09 μM) results from sequestration of

*Figure 8 continued on next page*

*Figure 8 continued*

the substrate by MLCP. In contrast, the IC$_{50}$ for dephosphorylation of pC-ERMAD is insensitive to its concentration because only one enzyme, MLCP, is primarily responsible and there is no sequestration.

The following figure supplements are available for figure 8:

**Figure supplement 1.** Dephosphorylation of pC-ERMAD by mouse uterus extracts.

**Figure supplement 2.** Determination of the 'strength' of competing phosphatase substrates and inhibitors in mouse uterus extracts.

(*Figure 8*). Based on previous measurements of the physiological MLCP concentration (*Alessi et al., 1992*; *Shirazi et al., 1994*) and accounting for extract dilution, the MLCP concentration in these extracts was ~0.08 µM. We expect that when the pCPI-17 concentration is in great excess (i.e., 1.35 µM in *Figure 8*), almost all its dephosphorylation will be due to PP2A-like enzymes rather than MLCP, given their much greater $k_{cat}$s. Because OA inhibits these PPU phosphatases with subnanomolar $K_i$s, the IC$_{50}$ is determined by, and allows us to compute, the PP2A-like PPU concentration in the extract [and by extension, the concentration of these enzymes in smooth muscle tissue (*Table 1* and *Supplementary file 1*)]. In contrast, when the pCPI-17 concentration is commensurate with, or lower than, the MLCP concentration (i.e. [pCPI-17]=0.09 µM or 0.01 µM, respectively), unfair competition predicts that most pCPI-17 will be sequestered by MLCP and that the IC$_{50}$ will be large because a high OA concentration is needed to compete with the tight pCPI-17/MLCP binding. The results shown in *Figure 8* conform to these predictions. In particular, note that the IC$_{50}$ for OA increases roughly 10× as the pCPI-17 concentration decreases from 1.35 µM to 0.01 µM because the identity of the relevant pCPI-17 phosphatase changes from PPU to MLCP.

The dephosphorylation of pC-ERMAD serves as a control in *Figure 8*. In the concentrated extracts, more than 75% of the activity against this substrate is due to PP1-containing enzymes [mostly, but not exclusively, MLCP; see *Figure 8—figure supplement 1* and (*Eto et al., 2000*; *Fukata et al., 1998*; *Liu et al., 2015*)]. Because the $K_m$ for the dephosphorylation of pC-ERMAD by MLCP (>2.5 µM; *Figure 2—figure supplement 1D* and *Table 2*) is much higher than the $K_i$ for OA (~20 nM; *Figure 4A* and *Table 2*), pC-ERMAD cannot compete effectively with OA for the active site of MLCP and other PP1-containing holoenzymes. We therefore anticipated that the OA IC$_{50}$ for the dephosphorylation of pC-ERMAD in concentrated uterus extracts should be unaffected by varying the substrate concentration. Again, the results shown in *Figure 8* concur with this prediction; the OA IC$_{50}$ is virtually identical when the pC-ERMAD concentration is 0.2 µM and when it is 2 µM.

The absolute rate of dephosphorylation at the lowest pCPI-17 concentration in the concentrated extract experiment provides another confirmation that MLCP, rather than other PPU phosphatases, dephosphorylates pCPI-17 when MLCP is in excess. As just explained, when pCPI-17 is at 0.01 µM virtually all it is sequestered by binding; therefore, the enzyme-substrate complex concentration is known (i.e. 0.01 µM) and we can determine the $k_{cat}$ of the responsible phosphatase to be ~0.007 s$^{-1}$. This result is in excellent agreement with ~0.008 s$^{-1}$, the mean $k_{cat}$ measured in vitro for purified recombinant MLCP (*Figures 2–5*) after adjustment for the 0°C reaction temperature that was required for technical reasons (see *Figure 7—figure supplement 2* and *Supplementary file 1*). The close fit between these two $k_{cat}$ values provides further support for the unfair competition model by identifying MLCP as the enzyme ultimately responsible for the dephosphorylation of sequestered pCPI-17 in concentrated extracts.

## pCPI-17 dephosphorylation by MLCP is crucial to timely MLCP reactivation

We next compared the unfair competition model with current ideas positing pCPI-17 dephosphorylation by PPU (*Eto and Brautigan, 2012*; *Eto et al., 2004*; *Hersch et al., 2004*; *Kitazawa, 2010*; *Obara et al., 2010*; *Takizawa et al., 2002*) in their abilities to explain the physiological experiments of *Kitazawa et al. (2009)* investigating the temporal responses of contracted rabbit femoral artery to sodium nitroprusside (SNP), a nitric oxide (NO) donor that induces vasodilation by partially deactivating the kinase(s) acting on pCPI-17. *Figure 9* shows their published measurements of the temporal changes in CPI-17 and myosin regulatory light-chain (MRLC) phosphorylation following

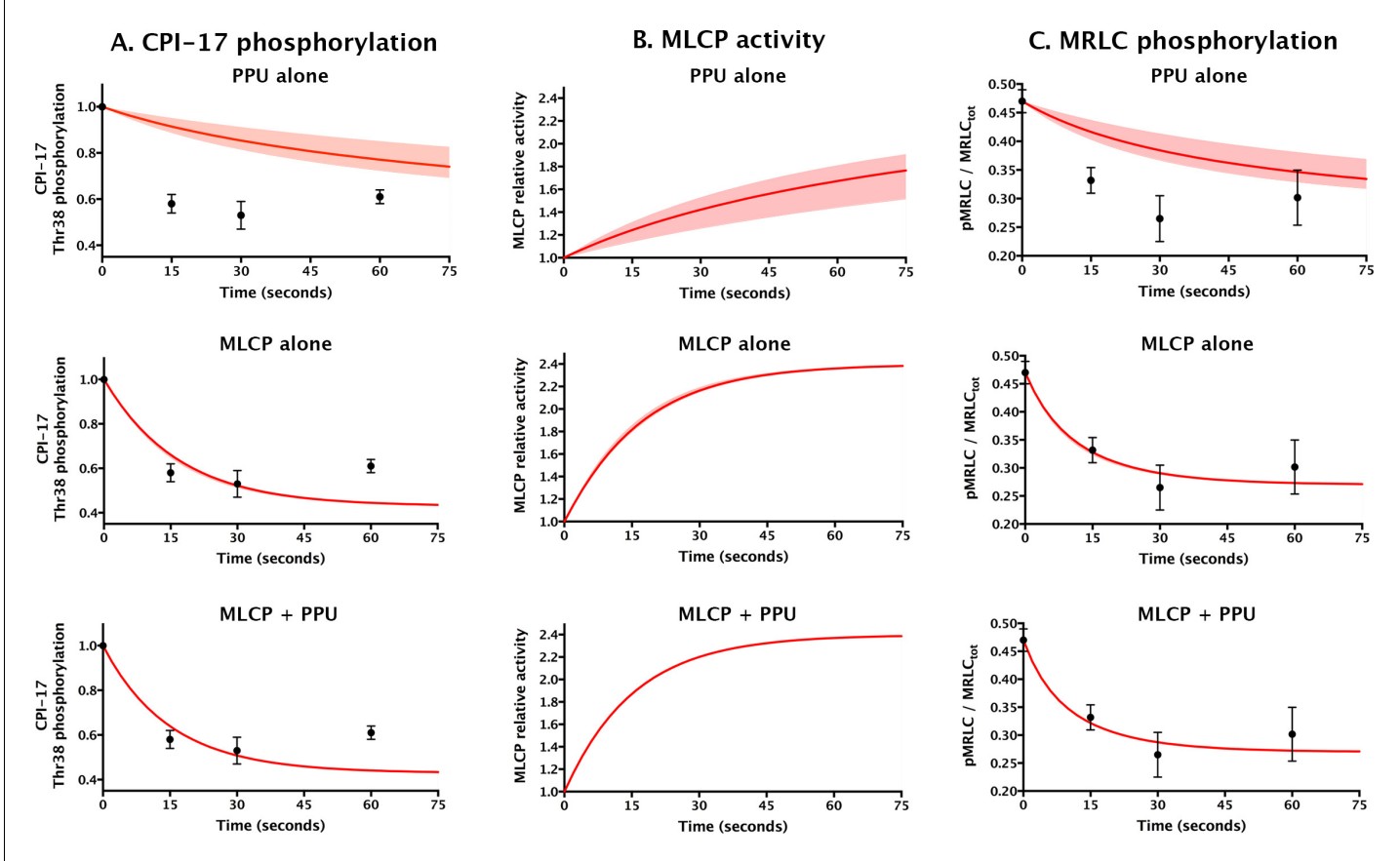

**Figure 9.** Comparing models for aorta relaxation upon application of vasodilator. Black points with error bars reproduce data from *Figure 4* in (*Kitazawa et al., 2009*) describing the temporal variation in the levels of (**A**) CPI-17 and (**C**) MRLC phosphorylation following the application of the vasodilator sodium nitroprusside to rabbit aorta at $t = 0$. Red lines show theoretical predictions (see *Supplementary file 1*) under three scenarios: top, PPU alone (current model, but allowing for sequestration of pCPI-17 by MLCP); middle, MLCP alone; and bottom, MLCP and PPU (unfair competition). (**B**) The computed relative MLCP activity on substrates other than pCPI-17. Shaded regions (which are barely discernable in the bottom two rows) indicate the variation in the predictions over the potential range of aortic MLCP concentration.

The following figure supplement is available for figure 9:

**Figure supplement 1.** Effect of ROCK thiophosphorylation on MLCP activity.

application of SNP to the tissue, along with the computational predictions under three scenarios for the phosphatases acting on pCPI-17: (i) PPU-alone (the current model, but accounting for the sequestration of pCPI-17 by MLCP), (ii) MLCP-alone, and (iii) both MLCP and PPU (unfair competition). To compute the simulations, we used the data from *Kitazawa et al. (2009)* to determine the pCPI-17-directed kinase and MLCP activities in the initial (vasoconstricted) and final (vasodilated) states, the pCPI-17/MLCP and pCPI-17/PPU parameters determined in this study (*Tables 1* and *2*), and the results from *Figure 8—figure supplement 2* that allowed us to account for the sequestration of both MLCP and PPU by other cellular substrates. From the change in MRLC phosphorylation, we inferred that MLCP activity increased ~2.4-fold following the addition of SNP, which is close to the ~3-fold increase measured by direct enzyme assays (*Etter et al., 2001*) in a system similar to that used by *Kitazawa et al. (2009)*. The shaded regions in *Figure 9* show the variation in the predictions of the various models over the potential range of MLCP concentrations (0.7–2 μM) determined from earlier studies in various smooth muscle tissues (*Alessi et al., 1992*; *Shirazi et al., 1994*). This variation is noticeable only in the PPU-alone model that ignores MLCP dephosphorylation of pCPI-17 (top row in *Figure 9*). In addition, varying the other relevant parameters over their

error ranges (*Table 1*) was found to have minor effect on any of the curves (not shown; see *Supplementary file 1*).

The major lesson of the modeling shown in *Figure 9* is that smooth muscle relaxation cannot take place with its observed rapid kinetics if MLCP does not participate in pCPI-17 dephosphorylation. The half-life for pCPI-17 dephosphorylation in the absence of MLCP activity (column A, top) is ~10× longer than the experimental ~10 s. As a result, the reactivation of MLCP is very slow (column B, top), as is the consequent dephosphorylation of MRLC (column C, top). The fit of this PPU-alone model to the published results is poor. On the other hand, an unfair competition system in which MLCP dephosphorylates pCPI-17 with the kinetic parameters we determined provides a good fit to the published data (bottom row). Interestingly, the rapidity of these events is virtually unaffected whether the system has no PPU activity or the substantial PPU activity indicated by our measurements (compare the middle and bottom rows in *Figure 9*). This outcome emerges because the contracted muscle starts out with MLCP in small excess of pCPI-17, so essentially all pCPI-17 is bound to MLCP, and the half-lives for pCPI-17 and MRLC dephosphorylations are determined almost exclusively by the $k_{cat}$ for the dephosphorylation of pCPI-17 by MLCP. That is, even though the tissue is well provisioned with PPU enzymes, they cannot access the MLCP-bound pCPI-17. We conclude that MLCP dephosphorylation of pCPI-17 is both necessary and sufficient for its reactivation during vasodilation.

## pCPI-17 dephosphorylation by MLCP is not importantly affected by physiological levels of MYPT1 inhibitory phosphorylation

The MYPT1 subunit of MLCP is subject to inhibitory phosphorylations at Thr[696] and Thr[853]. The phosphorylation at Thr[696] inhibits enzyme activity, while that at Thr[853] disrupts association of MLCP with its myosin substrate (*Khasnis et al., 2014*; *Velasco et al., 2002*). The kinase ROCK can phosphorylate both sites efficiently in vitro (*Feng et al., 1999a*, *1999b*; *Kawano et al., 1999*; *MacDonald et al., 2001*), but in vivo other kinases such as ZIPK, ILK, and PAK may also target pThr[696] [reviewed in (*Grassie et al., 2011*)]. In most smooth muscles, the levels of Thr[696] phosphorylation are high even in the relaxed condition, approaching 50% site occupancy. While most studies have concluded that pThr[696] levels are constitutive (*Chen et al., 2015*; *Kitazawa et al., 2003*; *Niiro et al., 2003*; *Tsai et al., 2014*; *Wilson et al., 2005*), some have reported modest increases in particular tissues stimulated to contract by certain agonists (*Bhetwal et al., 2013*; *Lubomirov et al., 2006*). One could thus argue that MLCP cannot be the enzyme that dephosphorylates pCPI-17 because MLCP in contracted tissue is inactivated by MYPT1 phosphorylation, particularly at Thr[696].

These considerations prompted us to determine whether inhibition of MLCP by Thr[696] phosphorylation might significantly affect the physiological process modeled in *Figure 9*. To this end, we thiophosphorylated MLCP with a molar excess of ROCK, and assayed the effects on pCPI-17 dephosphorylation (*Figure 9—figure supplement 1*). Because the amounts of MLCP used were too small to obtain reliable direct measurements of its thiophosphorylation level, we confirmed that our treatment inhibited its activity to at least the physiological extent using pC-ERMAD as a control substrate. ROCK phosphorylation of MLCP reduced MLCP's dephosphorylation of pC-ERMAD by ~70%, which is essentially the same as the maximum decrease in MLCP phosphatase activity observed in previous in vitro studies (*Feng et al., 1999a*, *1999b*; *Ichikawa et al., 1996*; *Khasnis et al., 2014*; *Murányi et al., 2005*, *Murányi et al., 2002*; *Wooldridge et al., 2004*). More importantly, this control establishes that in our experiments, a larger fraction of MLCP was inhibited by Thr[696] (thio)phosphorylation than in the physiological situation. Therefore, finding that the same preparations of thio (p)Thr[696] MLCP displayed activities against pCPI-17 that were reduced only ~20% relative to the untreated enzyme (*Figure 9—figure supplement 1*) provides an upper bound on the physiological inhibition of the MLCP-mediated pCPI-17 dephosphorylation rate. This small level of inhibition does not significantly affect the physiological modeling of *Figure 9*.

## Discussion

### MLCP is the enzyme primarily responsible for physiologically significant pCPI-17 dephosphorylation

Previous studies concluded that the dephosphorylation of pCPI-17 is accomplished by phosphatases other than MLCP [other PP1 holoenzymes, PP2A, and PP2C (*Eto and Brautigan, 2012*; *Eto et al., 2004*; *Hersch et al., 2004*; *Kitazawa, 2010*; *Obara et al., 2010*; *Takizawa et al., 2002*)] that can react with pCPI-17 much faster than MLCP. However, these studies failed to account adequately for the shielding of pCPI-17 from PPU enzymes when MLCP and pCPI-17 are present at physiological protein concentrations. This shielding results from the extremely low $K_m$ (0.48 ± 0.03 nM) of the pCPI-17/MLCP interaction.

This very low $K_m$ reflects the combined effects of a very high second-order association rate $k_{on}$ that must be in excess of $1.25 \times 10^8$ M$^{-1}$ s$^{-1}$, and slow unbinding ($k_{off}$) and catalytic ($k_{cat}$) rates. Such high association rates have been observed previously but are quite rare (*Archakov et al., 2003*; *Moal and Bates, 2012*). The high $k_{on}$ likely benefits from electrostatic MLCP-pThr[38] interactions (*Eto et al., 2007*; *Schreiber et al., 2009*). Moreover, $k_{on}$ is further increased while $k_{off}$ is decreased by the structural rigidity of pCPI-17 that results from its phosphorylation at Thr[38] (*Eto et al., 2007*). This rigidity increases the fraction of ligands that are in the bound conformational ensemble even before binding, and therefore reduces the loss of conformational entropy upon binding and the gain upon unbinding (*Archakov et al., 2003*; *Moal and Bates, 2012*). In addition, the structural rigidity of pCPI-17 inhibits its hydrolysis (*Eto et al., 2007*), accounting for the low $k_{cat}$ and decreasing the $K_m$ further [since $K_m = (k_{off} + k_{cat})/k_{on}$]. Consistent with the importance of pCPI-17 rigidity, point mutation of residues nearby Thr[38] that should result in increased flexibility coordinately increase both the IC$_{50}$ (up to 5000×) of MLCP inhibition and the $k_{cat}$ of pCPI-17 dephosphorylation (up to ~7.5 s$^{-1}$) (*Hayashi et al., 2001*).

### Unfair competition is necessary and sufficient for highly responsive regulation of CPI-17 phosphorylation

Applied to smooth muscle contraction/relaxation, our hypothesis of unfair competition posits that MLCP and pCPI-17 mutually sequester each other from other molecules during contraction, when sufficient CPI-17-directed kinase activity exists to maintain enough pCPI-17 to bind most MLCP. When the rate of phosphorylation decreases during relaxation (due to downregulation of PKC and other kinases targeting the pThr[38] site of CPI-17), MLCP is able to restart its own action against its normal substrates by dephosphorylating pCPI-17.

We demonstrated the various facets of this hypothesis in vitro using purified components (*Figures 2–6*), using crude extracts (*Figure 8*), and by in silico modeling (*Figures 5* and *9*). We showed that unfair competition is necessary for timely reactivation of myosin phosphatase because mechanisms that depend solely on the dissociation of pCPI-17 from MLCP would be far too slow (*Figure 9*). In fact, the modeling in *Figure 9* substantially underestimates the importance of unfair competition for the rapidity of smooth muscle relaxation because the MYPT1 regulatory subunit tethers the phosphatase to myosin filaments [reviewed by (*Khasnis et al., 2014*)]. In the vicinity of these filaments, the ratio of MLCP relative to PPU enzymes would be much higher than the cellular average used in the calculations.

The total CPI-17 concentration is large [~7 μM (*Kitazawa and Kitazawa, 2012*; *Woodsome et al., 2001*)], but our calculations using the data from *Kitazawa et al. (2009)* imply that only the relatively small fraction that is bound to MLCP [~1 μM (*Alessi et al., 1992*; *Shibata et al., 2015*)] is phosphorylated. This conclusion is supported by recent experimental studies. First, only a fraction of total CPI-17 is phosphorylated even in contracted arteries; the amount of pCPI-17 roughly corresponds to the concentration of MLCP (*Kitazawa and Kitazawa, 2012*). Second, in mice whose arteries contain a conditional knockout of MYPT1, the levels of pCPI-17 present after norepinephrine stimulation are considerably lower than those in stimulated wild-type animals, even though the total concentration of CPI-17 is unchanged (*Qiao et al., 2014*; *Tsai et al., 2014*). Our interpretation of this latter result is that the MYPT1 knockout exposed pCPI-17 to other phosphatases (PPUs) that dephosphorylated it rapidly.

We therefore envision that the role of PPU phosphatases in this mode of MLCP regulation is simply to ensure that cells always have little free pCPI-17 that is not bound by MLCP. The contractile apparatus thus can respond quickly and efficiently to reduction in the activities of CPI-17-targeting kinases such as PKC because, upon addition of a vasodilator, each MLCP molecule needs only to dephosphorylate the pCPI-17 molecule with which it is already associated. This idea explains why the speeds of pCPI-17 dephosphorylation and MLCP reactivation predicted by our model are virtually unaffected by PPU activity during the transition to vasodilation (*Figure 9*, middle and bottom rows). A curious corollary of this concept is that MLCP, the enzyme inhibited by pCPI-17, is ultimately responsible for protecting its inhibitor from the action of other phosphatases. If MLCP did not have this protective function, the intracellular concentration of pCPI-17 even during contraction would likely be too low to be of much significance in determining muscle tension.

If only the small fraction of CPI-17 that is phosphorylated is of relevance to MLCP inhibition, why do smooth muscle cells have such high levels of total CPI-17 [~7 μM (*Kitazawa and Kitazawa, 2012*; *Woodsome et al., 2001*)]? We suggest that high total CPI-17 may be needed to maintain even the small amount of pCPI-17 needed for inhibition. The CPI-17 kinases (e.g. PKC and ROCK) need to generate pCPI-17 at a rate sufficient to counteract the pCPI-17 phosphatases (MLCP and PPU). A requirement for high total CPI-17 may be particularly acute because most CPI-17 is cytosolic, while activated PKC and ROCK go to the plasma membrane upon activation (*Gong et al., 1997*; *Haller et al., 1990*; *Woodsome et al., 2001*).

## Downregulation of CPI-17-directed kinases is likely sufficient to explain the rapid phase of vasodilation

The unfair competition model quantitatively explains SNP's near-term (within 30 s) experimentally measured effects on CPI-17 and MRLC phosphorylation (*Kitazawa et al., 2009*) as consequences of a single event: an approximately 50% decrease in the activity of the kinases targeting CPI-17 Thr$^{38}$. Although we do not know of studies directly measuring the activities of these enzymes in response to vasodilator administration, the same report established the importance of their downregulation under the conditions employed, because pCPI-17 levels did not decrease after SNP addition in the presence of phorbol esters that artificially 'clamp' the activities of these kinases at high levels (*Kitazawa et al., 2009*).

The major kinases responsible for Thr$^{38}$ phosphorylation are Ca$^{2+}$-dependent PKC, Ca$^{2+}$-independent PKC (*Eto et al., 2001*), and Rho-associated kinase [ROCK; (*Kitazawa et al., 2000*)]. We believe that inactivation of Ca$^{2+}$-dependent PKC is of greatest consequence here because a straightforward mechanism could cut its activity significantly within a few seconds [reviewed by (*Kitazawa et al., 2009*; *Somlyo and Somlyo, 2003*)]. Nitric oxide (NO) released from SNP stimulates guanylyl cyclase to synthesize cGMP, which activates Protein kinase G (PKG). Phosphorylation of various PKG targets indirectly lowers levels of cytoplasmic Ca$^{2+}$, a cofactor for several forms of PKC. Because Ca$^{2+}$ binds PKC cooperatively (the Hill coefficient is between 2 and more than 8, depending on conditions), small decreases in the levels of Ca$^{2+}$ can potentially be amplified into much larger decreases of PKC activity (*Egea-Jiménez et al., 2013*). PKG can also downregulate ROCK, the other major CPI-17 Thr$^{38}$ kinase, by phosphorylating and thus inactivating RhoA (*Sauzeau et al., 2000*), although these changes in ROCK appear to be too slow to contribute much to the near-term events studied here (*Kitazawa et al., 2009*).

Some previous studies [e.g. (*Bonnevier and Arner, 2004*)] have posited that the addition of vasodilators such as SNP not only rapidly downregulates the kinases targeting CPI-17, but also upregulates one or more countering PPU phosphatases. Our identification of MLCP as the key physiological phosphatase for pCPI-17 brings this supposition into question. First, as discussed above, a PPU phosphatase would need to have a $K_m$ in the nanomolar or lower range to be able to compete with MLCP. Second, our quantitative analysis shows that a two-fold downregulation of CPI-17-targeting kinases following SNP addition is sufficient to explain the rapid loss of CPI-17 phosphorylation due to dephosphorylation by MLCP (see *Supplementary file 1* for details). The predicted half-life of this process is $t_{1/2} = \log 2/k_{cat} \approx 12$ s, which is in excellent agreement with experiment (*Kitazawa et al., 2009*). Any additional processes that significantly increased this rate of CPI-17 dephosphorylation would not be consistent with the physiological observations.

Although we doubt that NO-induced vasodilation involves significant increases in PPU enzyme activities, we cannot exclude that events other than pCPI-17 dephosphorylation might help

upregulate MLCP. One candidate is the phosphorylation of MYPT1 by PKG, which has been observed to increase considerably during the first minute after exposure to SNP (*Kitazawa et al., 2009*). PKG can phosphorylate MYPT1 at several sites, including Ser[668], Ser[692], Ser[695], and Ser[852] (*Wooldridge et al., 2004*; *Yuen et al., 2011*). The phosphorylations of two of these sites (Ser[695] and Ser[852]) appear not to affect MLCP activity directly, but instead prevent inhibitory Rho kinase-mediated phosphorylations (*Nakamura et al., 2007*; *Wooldridge et al., 2004*). However, after SNP treatment, these Rho kinase-mediated phosphorylations are removed too slowly to contribute to rapid MLCP activation within 15 s of SNP addition (*Kitazawa et al., 2009*). It remains possible that PKG-mediated phosphorylations of certain isoforms of MYPT1 at the other two sites (Ser[668] and/or Ser[852]) might participate in MLCP activation (*Yuen et al., 2014*), but our results suggest that any such contributions are relatively modest. On its own, the observed change in pCPI-17 levels after SNP administration can account for the increase in MLCP activity during vasodilation.

Concomitant with the rapid decrease in pCPI-17 levels and the rapid activation of MLCP after vasodilator addition is the key downstream physiological readout: the rapid decrease in myosin regulatory light chain (MRLC) phosphorylation (*Kitazawa et al., 2009*). In theory, lower levels of pMRLC could be achieved not only by upregulating MLCP, as detailed in this study, but also by inactivating myosin light chain kinase (MLCK). Given that SNP treatment reduces $Ca^{2+}$ levels and thus potentially can downregulate MLCK, it is perhaps surprising that the simulation in *Figure 9* was successful even though it assumed MLCK function was constant. Perhaps, in the experimental system of *Kitazawa et al. (2009)* analyzed in *Figure 9*, the SNP-induced decreases in the activity of CPI-17-targeting kinases were more important than any decrease in MLCK activity, which was likely at a very low basal level prior to SNP addition (*Dimopoulos et al., 2007*; *Isotani et al., 2004*). We note that even if MLCK inactivation plays a more significant role in determining MRLC phosphorylation during vasodilation than assumed by our model, no known pathway exists through which MLCK downregulation could impact the pCPI-17 levels or MLCP activities analyzed in *Figure 9*.

## Unfair competition may provide a general mechanism for the regulation of PPP phosphatases by AGC kinases

We previously showed that unfair competition is central to another important physiological situation: cell cycle transitions into and out of mitosis (*Williams et al., 2014*). During M phase, the PP2A-B55 phosphatase is inactivated by a mechanism involving small proteins of the Endosulfine/ARPP19 family (hereafter, Endos); these become PP2A-B55 inhibitors after their phosphorylation by the Greatwall kinase (Gwl) following the latter's activation by Cdk1-Cyclin B (*Blake-Hodek et al., 2012*; *Castilho et al., 2009*; *Gharbi-Ayachi et al., 2010*; *Mochida, 2014*; *Mochida et al., 2010*; *Williams et al., 2014*; *Yu et al., 2006*). Following inactivation of Cdk1-Cycliin B and Gwl at M phase exit, PP2A-B55 rapidly dephosphorylates pEndos and thus reactivates itself to attack mitosis-specific phosphosites. The dephosphorylation of pEndos by PP2A-B55 is both required and sufficient to account for the rapidity of pEndos dephosphorylation and of PP2A-B55 reactivation during M phase exit (*Williams et al., 2014*).

Unfair competition requires that the inhibitor/substrate bind very tightly to the active site and be dephosphorylated there only slowly. These requirements hold for the dephosphorylations both of pCPI-17 by MLCP (during muscle relaxation) and of pEndos by PP2A-B55 (during M phase exit); in fact, the kinetic constants are nearly identical in the two cases ($K_m$ = 0.48 ± 0.03 nM and 0.47 ± 0.014 nM, respectively; $k_{cat}$ = 0.06 ± 0.01 s$^{-1}$ and 0.02–0.07 s$^{-1}$, respectively). Of greatest interest, the $K_m$s are both subnanomolar; that is, lower by 2–3 orders of magnitude than that for any previously described dephosphorylation reaction.

Unfair competition provides an economical regulatory mechanism during changing cellular conditions: Only the upstream kinase that phosphorylates the inhibitor needs to be controlled; the sequel is an automatic consequence of the kinetic properties of the phosphatase to which the inhibitor is bound. No other steps that would introduce complexity to the system, such as allosteric control of a putative PPU enzyme, are needed.

Moreover, if such additional complexity is to be avoided, then unfair competition is likely the inevitable consequence of the well-known fact that PPP catalytic subunits can target a wide variety of substrates. Thus, in the cell, many molecules are competing for the catalytic sites, and we can estimate experimentally the magnitude of this competition (*Figure 8—figure supplement 2*; see *Supplementary file 1*). To inhibit an enzyme's ability to dephosphorylate a large number of

potential substrates, which will have a large aggregate concentration, a competitive inhibitor (e.g. pCPI-17 or pEndos) *must* have an extremely low $K_m$ for its target phosphatase (e.g. MLCP or PP2A-B55). Given the necessity for such very strong affinity between enzyme and inhibitor, inactivation of the inhibitor by the same enzyme provides the only pathway that allows cells and tissues to reset their state rapidly (e.g. during muscle relaxation or M phase exit).

These two examples of unfair competition involve processes with different fundamental requirements. Cell cycle transitions must operate much like on-off switches: Cells can be either in interphase or M phase, but cannot remain in an intermediate state. On the other hand, smooth muscle contractility must be regulated like a rheostat: Arteries often must be able to respond in a graded fashion to changing mixtures of agonists and vasodilators. How can phosphatase modules, both governed by unfair competition, participate in processes with such disparate needs? We suggest that the explanation lies in the presence (Endos, M phase) or absence (CPI-17, vasoconstriction) of positive feedback mechanisms that can convert a modest change in the activity of an upstream regulatory kinase into a discrete 'on-off' signal.

At the beginning of M phase, positive feedback increases Gwl activity by two pathways. The first mechanism is direct: Partially phosphorylated Gwl phosphorylates some pEndos which partially inhibits PP2A-B55; this further increases Gwl activity, presumably by decreasing the rate of Gwl dephosphorylation (*Mochida et al., 2016*). The other mechanism is indirect: pEndos inhibition of PP2A-B55 decreases the dephosphorylation of Cdc25 and Myt1/Wee1 by PP2A-B55, promoting the Cdk1 autoregulatory loop and consequently the activation of Gwl (*Blake-Hodek et al., 2012*; *Williams et al., 2014*; *Yu et al., 2006*) by raising the Gwl phosphorylation rate. Positive feedback also probably occurs at the end of M phase, when Gwl becomes dephosphorylated and turned 'off'. Although PP1 (*Heim et al., 2015*; *Ma et al., 2016*; *Mochida, 2015*; *Rogers et al., 2016*) and perhaps other enzymes (*Della Monica et al., 2015*) initiate Gwl inactivation, PP2A-B55 also likely plays a role in completing and then maintaining Gwl dephosphorylation (*Mochida et al., 2016*; *Yamamoto et al., 2011*).

In contrast, we know of no data indicating that pCPI-17 influences the activity of PKC or other upstream kinases that govern CPI-17 phosphorylation. Therefore, no feedback exists, consistent with our analysis in *Figure 9* indicating that CPI-17 kinases are not turned off completely even in the presence of vasodilators. Our calculations instead indicate that, under the experimental conditions, SNP-treated smooth muscles retain a basal CPI-17 kinase activity that is about one-half that of the contracted state (see *Supplementary file 1*). In other words, regulation of these kinases is graded rather than discontinuous.

It is striking that the kinases most likely to be involved in the phosphorylation of CPI-17 (PKC and ROCK) and Endos (Gwl) are all members of the AGC kinase family [reviewed in (*Arencibia et al., 2013*)]. This fact suggests that early in the evolution of the eukaryotic lineage, a prototypic AGC kinase phosphorylated an inhibitor of a prototypic PPP phosphatase and that the inhibitor and phosphatase interacted according to the logic of unfair competition. An intriguing speculation is that many of the other PPP holoenzymes that exist in eukaryotic cells eventually will be found to be regulated in similar ways by other AGC kinase family members.

## Materials and methods

### Cell lines

HEK293 CRL-1573 cells were obtained from the American Type Culture Collection (Manassas, VA). The identity of these cells was not subsequently authenticated because the cells were only used for the purification of reagents rather than as experimental materials. The cells were grown in Dulbecco's Modified Eagle Medium +10% Fetal Bovine Serum (Invitrogen, ThermoFisher Scientific) in the presence of 100 units/ml of penicillin and 100 μg/ml of streptomycin and a 1:100 dilution of the antimycotic Fungizone (Gibco, ThermoFisher Scientific).

### Purification of MLCP and other phosphatases

Human MYPT1 (PPP1R12A) cDNA was obtained from GE-Dharmacon (clone accession number BC111752). An N-terminal FLAG-peptide was fused to the MYPT1 coding sequence and cloned into the mammalian expression vector pcDNA5. pcDNA5-TO vector alone (Invitrogen), pcDNA5-TO/

FLAG-B56α, and pcDNA5-TO/FLAG-MYPT1 plasmid DNAs were transfected into HEK293 CRL-1573 cells using Lipofectamine 2000 (Invitrogen) using the manufacturer's protocols and harvested after 72 hr. Holoenzymes were isolated according to *Adams and Wadzinski (2007)*; *Williams et al., 2014*). Purified complexes were resolved by SDS-PAGE and visualized by silver staining. Protein concentrations were estimated by comparison to BSA standards. The compositions of the holoenzymes were verified by Western blot.

## Substrate purification and labeling

Human CPI-17 was cloned into a modified pGEX vector, creating an N-terminally 6×-His fusion. We used the Stratagene Quik-change kit to create a $Ser^{12}$-to-$Ala^{12}$ mutation in the recombinant CPI-17. The $Ala^{12}$ mutant was phosphorylated to ~80% of the wild-type level, suggesting that $Ser^{12}$ in wild-type CPI-17 was only minimally phosphorylated in vitro by the method that will be described below. In agreement, subsequent classical Michaelis-Menten experiments established that the $Ala^{12}$ mutant and the wild-type proteins were dephosphorylated by MLCP with the same $K_m$s and $k_{cat}$s, to within ~15% (n = 2). These two CPI-17 fusions (wild-type or the $Ala^{12}$ mutant) were subsequently transformed into *E. coli* BL21-Gold(DE3) cells (Agilent Technologies, catalog #230132). Freshly transformed cells were grown to an $OD_{600}$ of 0.6 in 2× yeast tryptone broth supplemented with 100 mg/ml of ampicillin, then induced with 1 mM isopropyl β-D-1-thiogalactopyranoside (IPTG) for 4 hr at 37°C.

Cells were harvested by centrifugation and stored at −70°C. Cells were lysed in 20 ml of lysis buffer (50 mM Tris-HCl, pH8.0; 500 mM NaCl; 5 mM imidazole, 5% glycerol, 0.025% Tween20, 5 mM β-mercaptoethanol) supplemented with Complete EDTA-free protease inhibitor (Roche) and with 300 µg/ml of lysozyme, followed by sonication to complete lysis. Cleared lysates were bound to Ni-NTA Agarose beads (Qiagen) overnight at 4°C, then transferred to a column and washed with 250 mL of wash buffer (10 mM Tris-HCl, pH 8.0; 500 mM NaCl; 15 mM imidazole, 40 mM glycine, 0.025% Tween20, 5 mM β-mercaptoethanol, and 5% glycerol). CPI-17 was then eluted in 10 mM Tris-HCl, pH 8.0; 500 mM NaCl, 200 mM imidazole, 0.025% Tween20, 5 mM β-mercaptoethanol, and 5% glycerol. Eluted protein was concentrated and exchanged into storage buffer (25 mM Tris-HCl, pH 8.0; 500 mM NaCl, 0.025% Tween20, 5 mM β-mercaptoethanol, and 5% glycerol) using Amicon Ultra-4 centrifugal filters with a 3 K molecular weight cutoff (Millipore).

Purified CPI-17 was diluted into PKC buffer (25 mM MOPS, pH 7.2; 12.5 mM β-glycerophosphate, 25 mM $MgCl_2$, 5 mM NaF, 0.25 mM DTT, 0.025% Tween20, 5 mM β-mercaptoethanol, and 5% glycerol) supplemented with PKC lipid activator (Millipore, catalog #20–133). CPI-17 was then phosphorylated in vitro by PKCδ (SignalChem, catalog #P64-10G) in the presence of 10–100 µM ATP and $γ$-$^{32}$P-ATP (50 µCi at 6000 Ci/mmol). Reactions were incubated at 30°C overnight. Proteins were purified away from reaction components and desalted using Micro Bio-Spin six columns (Bio-Rad) equilibrated with B3 buffer (25 mM Tris, pH 7.5; 150 mM NaCl; 0.01 µg/ml bovine serum albumin; 0.05% Tween20; 5 mM β-mercaptoethanol, 5% glycerol). For some studies, the CPI-17 protein was thiophosphorylated by PKCδ in the presence of 1 mM [$γ$-$^{35}$S]thio-ATP. Control experiments showed that purified MLCP did not detectably dephosphorylate [$γ$-$^{35}$S]thio(p)CPI-17.

Human myosin regulatory light chain (MRLC, also called MYL9) cDNA was obtained from GE-Dharmacon (clone accession number BC002648) and used to create an N-terminal maltose binding protein fusion in the vector pMAL-C2X (New England Biolabs). MBP-tagged MRLC was expressed in *E. coli* as previously described (*Castilho et al., 2009*; *Williams et al., 2014*). For $^{32}$P-labeling by myosin light-chain kinase [MLCK (1425–1776); Sigma-Aldrich catalog #M9197] purified MRLC was diluted into MLCK buffer (20 mM MOPS, pH 7.2; 12.5 mM β-glycerophosphate, 25 mM $MgCl_2$, 5 mM EDTA, 2 mM EGTA, 5 mM NaF, 0.25 mM DTT, and 5% glycerol) in the presence of 5 mM $CaCl_2$, 0.0375 µg/µl calmodulin, 100 µM ATP and [$γ$-$^{32}$P]ATP (50 µCi of 6000 Ci/mmol). Reactions were incubated at 30°C overnight. MRLC was purified away from the kinase and other reaction components, eluted with glutathione, and desalted and concentrated using Amicon Ultra-4 10K centrifugal filters (Millipore) and buffer B3.

Ezrin GST-C-ERMAD recombinant protein produced in *E. coli* and LOK kinase produced in HEK293 CRL-1573 cells were purified according to *Viswanatha et al. (2014)*. GST-C-ERMAD bound to glutathione agarose beads was labelled with LOK kinase in the presence of [$γ$-$^{32}$P]ATP, and eluted with glutathione according to protocols of *Viswanatha et al. (2014)*.

Bovine myelin basic protein (MyBP) was from Millipore and was [32]P-phosphorylated in vitro by Protein Kinase A (New England Biolabs) using methods previously described (*Castilho et al., 2009*; *Williams et al., 2014*).

## Phosphatase assays with purified components

Purified holoenzymes were diluted to the appropriate concentrations in reaction buffer (20 mM Tris, pH 7.5; 150 mM NaCl), while substrates and inhibitors were diluted in buffer B3; see figure legends for specific concentrations and conditions. In general, 6 µl phosphatase assays were made up of 2 µl enzymes, 2 µl of substrate, and 2 µl of buffer/inhibitors. Reactions containing no enzyme were included as controls to demonstrate that the substrates did not contain phosphatase activity. A time course pilot experiment was performed for each extract/substrate combination to ensure that the assays were performed in the linear range (*Figure 2—figure supplement 1*, *Figure 6* and *Figure 7— figure supplement 2*). All experiments investigating rates of pCPI-17 dephosphorylation were performed with the Ala[12] mutation described above to ensure more precise quantitation. For okadaic acid (OA) and calyculin A inhibition experiments (*Figure 4*), the inhibitors were serially diluted in DMSO, and then mixed with MLCP and incubated 10 min on ice before the addition of substrates. Phosphatase assays were stopped with trichloracetic acid (TCA) after a time at which <20% (typically in the range of 5–10%) of the substrate was dephosphorylated. The supernatant was mixed with ammonium molybdate and extracted with heptane-isobutyl alcohol according to *Mochida and Hunt, 2007*. [32]P release was then measured using a PerkinElmer liquid scintillation counter (Tri-Carb 2810TR). Data were analyzed and statistics were generated by non-linear regression to Michaelis-Menten kinetics using Prism 6 software (GraphPad Software).

For the experiments detailed in *Figure 9—figure supplement 1*, MLCP was first thiophosphorylated by ROCK-II (Millipore, catalog #11–552) as follows. MLCP (0.5 nM) and ROCK-II (2.5 nM) were incubated at 30°C for 90 min in the presence of 0.1 mM γ-thio-ATP, as described in *Khasnis et al. (2014)*. After thiophosphorylation, standard phosphatase assays were performed as detailed in the figure legend.

## Phosphatase assays in cell extracts

Mouse uteri were dissected in phosphate-buffered saline (PBS) and immediately frozen in liquid nitrogen until use. Uteri were homogenized in ice-cold TPER buffer (ThermoFisher Scientific) supplemented with 1× Halt protease inhibitor (ThermoFisher Scientific), 1x Protease inhibitor tablet (Pierce), 0.1 mg/ml Soybean Trypsin Inhibitor (Sigma), and 1 mM phenylmethylsulfonylfluoride (PMSF; Sigma). This combination of protease inhibitors was essential for prevention of enzyme and substrate degradation in the extract. After centrifugation, the extract supernatant was assayed for phosphatase activity against the substrates described above. A typical assay would be composed of 3 µl of extract mixed with 1 µl of substrate. For OA titration, the drug was serially diluted in DMSO and mixed 1/10 in the extract, which was then incubated 10 min on ice before addition of substrate. For kinetic studies, [[32]P]pCPI-17 was added at concentrations detailed in the figure legends. Phosphatase assays were performed as described previously (*Castilho et al., 2009*; *Williams et al., 2014*).

## Acknowledgements

This work was supported by National Institute of Health grant GM048430 to M.L.G.

## Additional information

### Funding

| Funder | Grant reference number | Author |
| --- | --- | --- |
| NIH Office of the Director | GM048430 | Michael L Goldberg |

The funders had no role in study design, data collection and interpretation, or the decision to submit the work for publication.

## Author contributions
JJF, Data curation, Formal analysis, Investigation, Methodology; BCW, Conceptualization, Data curation, Formal analysis, Investigation, Methodology; ME, Conceptualization, Resources, Visualization, Writing—review and editing; DS, Conceptualization, Formal analysis, Methodology, Writing—original draft; MLG, Conceptualization, Formal analysis, Supervision, Funding acquisition, Methodology, Writing—original draft, Project administration

## Author ORCIDs
Michael L Goldberg, http://orcid.org/0000-0003-0200-0277

## Additional files

### Supplementary files
• Supplementary file 1. Theoretical methods.

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
