## [Decision Letter]

Thank you for submitting your article "Unfair Competition Governs the Interaction of pCPI-17 With Myosin Phosphatase (PP1-MYPT1)" for consideration by *eLife*. Your article has been favorably evaluated by Jonathan Cooper (Senior Editor) and three reviewers, one of whom is a member of our Board of Reviewing Editors. The following individuals involved in review of your submission have agreed to reveal their identity: Béla Novák (Reviewer #2) and Matthieu Bollen (Reviewer #3).

The reviewers have discussed the reviews with one another and the Reviewing Editor has drafted this decision to help you prepare a revised submission.

Summary:

Protein phosphatase PP1-MYPT1 is inhibited by a phosphorylated form of CPI-17 which has previously been proposed by various groups to act as a pseudosubstrate. In this manuscript it is demonstrated that pCPI-17 is actually a real substrate of PP1-MYPT1 that binds with an extremely high affinity but is dephosphorylated very slowly. This prevents the phosphatase from dephosphorylating other substrates as long as stoichiometric amounts of pCPI-17 are present, but also protects pCPI-17 from dephosphorylation by other phosphatases. The described regulation of PP1-MYPT1 by pCPI-17 through 'unfair competition' is similar to that of PP2A-B55 by p-endosulfine, which was previously reported by the same group. The data are interesting and convincing. However, as detailed below, the manuscript lacks some key information. Also, repetitive elements (both illustrations and text) should be removed.

Essential revisions:

1) The manuscript is too long. In particular, the Introduction and Discussion sections are repetitive and not entirely to the point, and can improved by substantially reduced length.

2) The beginning of the manuscript Introduction should provide basic information concerning the biology of muscle relaxation and the role of protein phosphorylation for the non-expert.

3) The generality of "unfair competition" stated in the Abstract makes the current study appear to be an incremental advance. This is a good discussion point, but the inclusion of this information in the Abstract diminishes the study.

4) Figure 3, and Figure 7 show basically the same. One of the two panels of each figure could be shown as a supplement.

5) Figure 10 is difficult to interpret since C-ERMAD was used at a concentration far below its Km while CPI-17 was used at a concentration far above its *K_m_*. This experiment should be repeated at different concentrations of each substrate so that the effect of ROCK on the kinetic parameters can be deduced.

6) A molecular explanation for the slow dephosphorylation rate of CPI-17 by PP1-MYPT1 is lacking and not discussed.

7) CPI-17 exists in a 7-fold molar excess to PP1-MYPT1 (Table 1). The proposed model does not sufficiently take into account the need and significance of this molar excess.

---

## [Author Response]

*Essential revisions:*

*1) The manuscript is too long. In particular, the Introduction and Discussion sections are repetitive and not entirely to the point, and can improved by substantially reduced length.*

*2) The beginning of the manuscript Introduction should provide basic information concerning the biology of muscle relaxation and the role of protein phosphorylation for the non-expert.*

*3) The generality of "unfair competition" stated in the Abstract makes the current study appear to be an incremental advance. This is a good discussion point, but the inclusion of this information in the Abstract diminishes the study.*

*4) Figure 3, and Figure 7 show basically the same. One of the two panels of each figure could be shown as a supplement.*

The reviewers felt that the manuscript was too long (#1) and lost focus by raising the issue of the generality of unfair competition in the Abstract (#3). They also requested that we add: information to the Introduction concerning the biology of muscle relaxation and the role of protein phosphorylation in this process (#2); a new section of the Discussion providing a molecular explanation for the unusual kinetics of the pCPI-17/MLCP interaction (#6); and a new section of the Discussion explaining the need for a 7-fold molar excess of total CPI-17 to MLCP (#7). The reviewers asked us to split out panels of Figure 3 and Figure 7 as supplementary figures (#4).

In response, we cut roughly 20% of the material from the original Introduction and Discussion. After addition of the new information requested, the revised Introduction is ~8% shorter than the original, and the revised Discussion is ~12% shorter. We removed mention of the generality of unfair competition from the Abstract and also split out the two figures as requested. We feel that these changes strengthened the paper and thank the reviewers for their suggestions.

*5) Figure 10 is difficult to interpret since C-ERMAD was used at a concentration far below its Km while CPI-17 was used at a concentration far above its K_m_. This experiment should be repeated at different concentrations of each substrate so that the effect of ROCK on the kinetic parameters can be deduced.*

Here, the reviewers asked us to repeat and extend the experiments that had been reported in Figure 10 of the original submission so as to deduce the kinetic parameters of the interaction between pCPI-17 and ROCK-thiophosphorylated MLCP. Although this issue was not a central component of our work, it was indeed a weakness of the previous manuscript, and the reviewers’ comment impelled us to review the literature and our assumptions about the effects of the inhibitory phosphorylations on MLCP.

Our interpretation of our original data was that the Thr^696^ phosphosite on MYPT1 (a subunit of MLCP) occupies the enzyme’s active site and thus works as a competitive inhibitor; thus, pCPI-17 with its extraordinarily low *K_m_* would be less susceptible to this kind of inhibition than other substrates. An alternative explanation is that the MYPT1 phosphorylation is allosteric in a manner that affects the interaction of the phosphatase with pCPI-17 less than with other substrates. In fact, the literature on this point is confusing, with some evidence for both points of view.

After extensive consideration, we feel that we can in fact contribute very little to this topic because the amounts of MLCP we can purify are too small to quantitate the degree of MYPT1 phosphorylation; this would be essential to discriminating between models. And ultimately, this issue is beside the point of our manuscript. What we wanted most to do here was to determine whether this MYPT1 phosphorylation would negate our modeling in Figure 9 by slowing down substantially the action of MLCP on pCPI-17. We are confident that our results show that this is not the case. Our controls show that we are disrupting ~70% of the enzyme’s function (for whatever reason), which is a higher degree of inhibition than has ever been seen in vivo. Yet the dephosphorylation of pCPI-17 is inhibited only by 20%, making this an upper limit on the degree to which this mechanism would slow up the dephosphorylation of pCPI-17 by MLCP. Our models would be virtually unaffected by this small effect.

Therefore, we have not performed the requested experiment but instead changed our presentation of the results already obtained. We no longer use our data to distinguish the molecular mechanisms by which phosphorylations of MYPT1 affect MLCP activity, but instead mention only that these experiments show this phosphorylation can influence pCPI-17 dephosphorylation by MLCP only marginally. Much discussion has been removed and the figure has been downgraded to a supplement.

*6) A molecular explanation for the slow dephosphorylation rate of CPI-17 by PP1-MYPT1 is lacking and not discussed.*

See response to comments #1-4.

*7) CPI-17 exists in a 7-fold molar excess to PP1-MYPT1 (Table 1). The proposed model does not sufficiently take into account the need and significance of this molar excess.*

See response to comments #1-4.